# PATIENTSIM: A Persona-Driven Simulator for Realistic Doctor-Patient Interactions

**Daeun Kyung**[1], **Hyunseung Chung**[1], **Seongsu Bae**[1], **Jiho Kim**[1],
**Jae Ho Sohn**[2], **Taerim Kim**[3], **Soo Kyung Kim**[4,*] **Edward Choi**[1,*]
[1]KAIST  [2]UCSF  [3]Samsung Medical Center  [4]Ewha Womans University
{kyungdaeun,edwardchoi}@kaist.ac.kr, sookim@ewha.ac.kr

## Abstract

Doctor-patient consultations require multi-turn, context-aware communication tailored to diverse patient personas. Training or evaluating doctor LLMs in such settings requires realistic patient interaction systems. However, existing simulators often fail to reflect the full range of personas seen in clinical practice. To address this, we introduce PATIENTSIM, a patient simulator that generates realistic and diverse patient personas for clinical scenarios, grounded in medical expertise. PATIENTSIM operates using: 1) clinical profiles, including symptoms and medical history, derived from real-world data in the MIMIC-ED and MIMIC-IV datasets, and 2) personas defined by four axes: personality, language proficiency, medical history recall level, and cognitive confusion level, resulting in 37 unique combinations. We evaluate eight LLMs for factual accuracy and persona consistency. The top-performing open-source model, Llama 3.3 70B, is validated by four clinicians to confirm the robustness of our framework. As an open-source, customizable platform, PATIENTSIM provides a reproducible and scalable solution that can be customized for specific training needs. Offering a privacy-compliant environment, it serves as a robust testbed for evaluating medical dialogue systems across diverse patient presentations and shows promise as an educational tool for healthcare. The code is available at `https://github.com/dek924/PatientSim`.

## 1 Introduction

Large language models (LLMs) have shown impressive performance on medical question-answering benchmarks such as MedQA [21], MedMCQA [44], and PubMedQA [22], even surpassing human experts. However, these benchmarks use single-turn settings where patient data is readily provided, and models simply analyze these data to select the most likely diagnosis or treatment. In contrast, real-world clinicians engage in multi-turn, context-aware conversations to gather patient information actively. As a result, these models may not guarantee effectiveness in practical clinical settings.

To evaluate LLM-powered virtual doctors (*i.e.*, doctor LLMs) in multi-turn settings, realistic patient interaction systems are needed. Traditionally, standardized patients (SPs) [4], trained actors simulating symptoms and histories, have been used to train and assess medical students' communication and clinical skills. In this context, SPs could serve as a benchmark for evaluating doctor LLMs by providing dynamic, interactive patient encounters. However, SPs are limited by high costs, inconsistent availability, and scaling challenges due to the need for human actors [13]. In contrast, LLM-based patient simulators provide a scalable, accessible, and cost-effective alternative [8]. They reduce the need for repetitive human acting, eliminate geographic and time constraints, and lower costs compared to SPs. These advantages highlight the potential of AI as a powerful tool for training and evaluating medical students [19, 34, 61], as well as doctor LLMs [14, 26, 32, 33, 35, 51, 58].

---

*Co-corresponding author

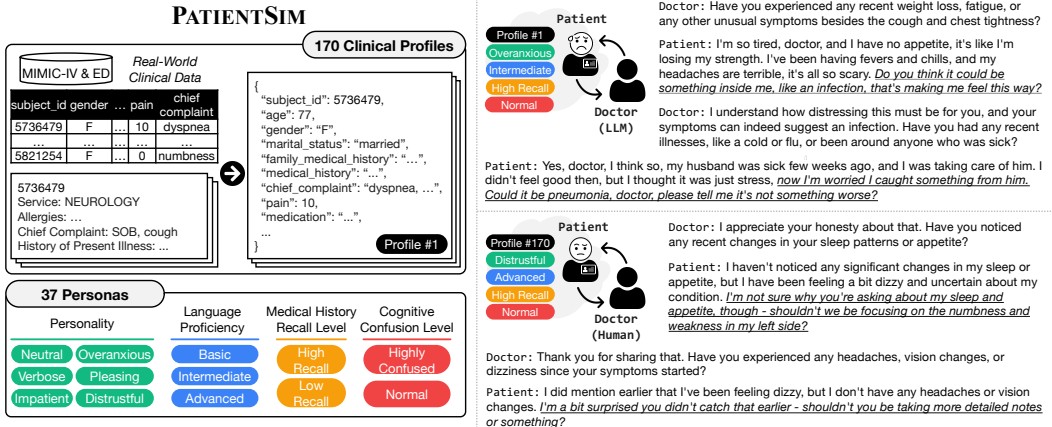

Figure 1: Overall framework of PATIENTSIM. PATIENTSIM generates realistic doctor-patient conversation data based on 1) clinical profiles, including symptoms and medical history, derived from MIMIC-ED and MIMIC-IV datasets, and 2) personas defined by four axes: personality, language proficiency, medical history recall level, and cognitive confusion level, resulting in 37 unique combinations (left). Each simulation begins with a doctor's question, where the doctor only has access to the patient's basic information, not their symptoms. The examples shown are mid-dialogue snippets selected to highlight the patient's persona (right).

Recent work highlights the potential of LLM-based patient simulators, but a significant gap remains between these systems and real clinical settings. A number of studies [33, 35, 38, 51] explored doctors' interactive information-seeking abilities by providing LLMs with patient data and having them role-play patients. However, these studies focused on evaluating the performance of doctor LLMs, even though the validity of these evaluations depends on how closely patient simulators emulate actual patient behavior. Recognizing this importance, some studies [12, 14, 36] have begun evaluating patient simulators focusing on how accurately they convey symptomatic information. However, doctor-patient consultations are more than just patients accurately reciting their symptoms. *Effective consultations must take into account patient behaviors dictated by multiple axes such as their emotional states and language skills, which significantly influence health outcomes.*

To this end, we propose PATIENTSIM, a system that simulates diverse patient personas encountered in clinical settings (Figure 1). Our simulator acts based on: 1) clinical profiles, including symptoms and medical history, and 2) personas defined by four axes: personality, language proficiency, medical history recall level, and cognitive confusion level. Patient profiles are constructed based on real-world medical records from the MIMIC-ED [23] and MIMIC-IV [25] datasets, totaling 170 profiles (Sec. 4). For personas, we defined 37 distinct combinations across four axes, designed to reflect key factors impacting doctor-patient consultation quality, based on literature reviews and guided by medical experts (Sec. 5.1). We evaluate eight LLMs as the backbone of our simulator and select Llama 3.3 70B as the final model, which maintains a persistent persona while ensuring factual accuracy (Sec. 7). The resulting simulator was assessed by four clinical experts and received an average quality score of 3.89 out of 4 across six criteria. Our simulator offers the following contributions:

- PATIENTSIM introduces a novel framework for simulating realistic doctor-patient interactions. It leverages real-world clinical data from MIMIC-IV and MIMIC-ED, modeling diverse patient personas across four axes: personality, language proficiency, medical history recall level, and cognitive confusion level.

- We conduct a comprehensive evaluation across eight LLMs, assessing factual accuracy and persona reflection. To confirm the robustness of PATIENTSIM 's simulations, the top-performing open-source model, Llama 3.3 70B, is further validated by four clinicians.

- Built on an open-source model, PATIENTSIM offers an accessible, reproducible tool for providing doctor-patient consultation data while prioritizing patient privacy. This scalable, privacy-compliant solution enables researchers and practitioners to validate their model's performance and adapt it for clinical uses such as educational tools.

## 2    Related work

**LLM-based agent simulation in clinical setting**    LLM-based agent simulations in clinical set-tings vary by scope and agents. Previous studies [1, 3, 32, 62] simulate hospital workflows with agents such as patients, nurses, and physicians, prioritizing final task accuracy (*e.g.*, diagnositic or department recommendation accuracy) over agent interactions. Additionally, previous works such as Medagents [55] and MDAgents [29] focus on collaborative physician decision-making but are limited to single-turn QA settings. Recent studies emphasize doctor-patient interactions, evaluating physician LLMs in patient-centered communication [1, 36, 38, 51] or exploring their potential as educational tools [19, 34, 61]. However, these often overlook diverse patient characteristics, leading to insufficient realism in simulated interactions. As patient simulators are foundational to hospital simulations, providing primary clinical information and driving interaction dynamics, ensuring their realism is key challenge. Our research addresses this by developing an LLM-based patient simulator that delivers clinically coherent responses and reflects diverse patient characteristics.

**LLM patient simulation**    LLM-based patient simulation is divided into applications for general hospital consultations and psychological consultations, each with distinct objectives. In general hospital settings, patient simulations aim to accurately present medical history and symptoms through multi-turn dialogue [12, 14, 36, 38]. While most efforts primarily focus on implementing patient simulator that can respond to questions with factually correct answers, some studies [12, 38] tried to add a bit of realism to the patient simulator by describing its personas with keywords such as the Big Five traits or occupation groups. No previous studies, however, aimed to implement and, at the same time, evaluate patient simulator that can emulate diverse and clinically relevant personas. Psychological counseling simulations, on the other hand, try to model complex internal states, such as thoughts, and emotions, emphasizing subjective responses like mood shifts or treatment resistance [39, 47, 59, 60]. These studies therefore prioritize deeper persona development to capture emotional and relational nuances, making them unsuitable for simulating patients in general diagnostic consultation settings such as hospital emergency departments. Unlike previous works, our study proposes a realistic patient simulator that combines the emotional realism emphasized in psychological counseling with the clinical accuracy required for general diagnostic consultations.

## 3    Problem definition

We target a first-time, single-session emergency department (ED) visit, considering technical and clinical constraints below, to ensure a feasible and coherent design.

**Limited access to comprehensive clinical data**    In real-world clinical practice, physicians integrate data from multiple dimensions such as patient-reported symptoms, physical examinations, and test results (*e.g.*, laboratory or imaging studies) to make diagnoses. However, when simulating patients, it is infeasible to predefine or dynamically generate clinically accurate, case-consistent data across all relevant dimensions to address the full range of possible queries. Even when using real hospital data, available examinations and test results are frequently sparse or incomplete. Moreover, reliable methods to generate accurate synthetic test results to fill these gaps are still lacking. As a result, patient simulators often receive questions about data that are inaccessible. Prior approaches have addressed this by instructing LLMs to respond with vague statements like "I don't know" [12, 33, 36] or to assume normal test results when data are not defined in the patient's profile [14, 51]. These strategies restrict physicians to inquire predefined data, limiting their ability to explore diverse reasoning paths. Moreover, assuming normal test results can mislead physicians. To address this, we focus our simulation on the *history-taking* process, a systematic approach to gather patient's personal and medical information, prior to physical exam or lab tests [27]. Research shows that approximately 80% of diagnoses rely on history taking alone, highlighting its critical importance [18, 45]. Unlike objective data (*e.g.*, test results), subjective data (*e.g.*, patients' verbal reports) inherently involves uncertainty [41, 52], allowing it to capture the variability in patients' descriptions. This variability, driven by personal factors, supports our goal of creating realistic virtual patients with diverse personas, ensuring flexible and naturalistic interactions.

**Inability to simulate longitudinal patient state changes**    Realistic multi-session simulations require modeling treatment effects and disease progression over time. However, current methods are limited to specific cases and rely on restrictive assumptions. In our setting, where the doctor LLM is allowed

to ask open-ended questions freely, incorporating interventions, medications, or follow-up strategies presents a risk, as the doctor LLM may suggest treatments different from those in the database. Patient outcomes can vary substantially depending on numerous factors, such as comorbidities, lifestyle, and medication adherence. Predicting these trajectories is highly complex, even for experienced physicians. Attempting to model longitudinal clinical states with existing methods risks generating misleading or clinically unsafe conclusions. Therefore, we focus on single-session interactions, avoiding the need to model long-term outcomes or readmissions.

**Problem scope** In the initial ED consultation setting, physicians often rely on verbal patient information, such as symptoms and medical history, for differential diagnosis under time pressure, before test results become available. Thus, we focus on differential diagnosis based on this initial consultation, which typically does not require test data. This approach ensures clinical relevance by reflecting real-world diagnostic reasoning while sidestepping current technical limitations.

# 4 Patient profile construction

**Structurized patient profile** Simulating patients requires detailed clinical information, such as presenting symptoms, pain levels, and family medical history, which is essential for clinicians to perform clinical reasoning and decision-making during the initial diagnostic process [57]. Currently, the MIMIC database is the only publicly available resource that provides this level of clinical detail. To ensure clinical relevance while minimizing ambiguity in the simulations, we constructed detailed and structured patient profiles using real clinical data from MIMIC-IV [23], MIMIC-IV-ED [25], and MIMIC-IV-Note [24]. We extracted accurate patient data from structured tables and used clinical notes to capture detailed information, such as lifestyle and present symptoms, not included in the tables. This hybrid approach combines the accuracy of structured data with the depth of narrative notes. Details on data sources and processing are provided in the Appendix A.1 and A.2. As a result, each patient profile includes 24 items, covering demographics, social and medical history, and ED visit details (see Appendix A.3). Clinical experts reviewed each item for relevance.

**Target disease selection** We select the five prevalent diseases from the MIMIC-IV-ED dataset: myocardial infarction, pneumonia, urinary tract infection, intestinal obstruction, and cerebral infarction (stroke). Our simulator operates exclusively in a dialogue-based setting due to several technical and clinical constraints (Sec. 3). Within this scope, we select five diseases based on the following criteria: 1) high clinical prevalence and significance as common ED presentations; 2) distinct and differentiable symptom profiles suitable for diagnosis through history-taking alone; and 3) sufficient representation in the MIMIC-ED to support robust model development and evaluation. Since the target diseases must be reliably inferred from conversational information alone without any test results, the range and granularity of possible disease categories is limited. For example, physicians cannot reliably distinguish between myocardial infarction and angina solely through patient dialogue. The selection process was guided by two medical experts, one of whom is an ER doctor with 13 years of experience, to ensure clinical relevance and consistency with ED workflows.

# 5 PATIENTSIM

## 5.1 Persona definition

We defined four key axes for persona simulation that impact consultation quality in clinical practice, based on literature reviews and guidance from medical experts.

**Personality** Personality is a well-established factor influencing consultation quality [7, 9, 37, 49]. The Big Five framework [40], one of the most widely recognized models of personality, has been used in previous patient simulation studies [12], but its traits are broad and tend to influence patient–physician interactions only indirectly. Recent psychological therapy research emphasizes observable conversational styles that directly manifest in patient interactions. Drawing on this, we adapt these styles into doctor-patient consultation-specific personality that are directly observable and actionable for simulation. Based on literature review [2, 6, 31, 53] and guidance from medical experts, we define six personalities relevant to medical consultations in ED: impatient, overanxious, distrustful, overly positive, verbose, and neutral (straightforward communication) as the baseline.

**Language proficiency** A patient's language proficiency is a critical determinant of doctor-patient communication quality [46, 54], yet it has been underexplored in simulation contexts. By specifying language proficiency levels, we simulate scenarios in which physicians must adapt to patients with varying proficiency by using appropriate language to ensure understanding. We use the Common European Framework of Reference for Languages (CEFR) [42], which defines six proficiency levels (A1, A2, B1, B2, C1, C2). To facilitate the human evaluation by physicians, we consolidated these into three levels, A (basic), B (intermediate), and C (advanced).

**Medical history recall level** Patients may not always accurately recall the details of their medical history [5, 30]. Assuming perfect recall, as in traditional settings, represents an idealized case. In low-recall scenarios, physicians must ask additional questions to build diagnostic confidence. We define two settings: high recall and low recall, enabling practice with diverse patient profiles.

**Level of cognitive confusion** Patients visiting the ED often present acute symptom exacerbation, leading to a highly confused and dazed state. These patients may initially struggle with coherent communication but stabilize through interaction. To simulate such cases, we define two mental status levels: highly confused and normal.

To avoid overlap between confusion and other axes (*e.g.*, impatient personality, low language proficiency, or low recall), highly confused patients are limited to neutral personality, intermediate language proficiency, and high recall. A detailed analysis of the impact of high confusion on other persona traits is provided in Appendix F.4.3. This results in 37 distinct personas; 36 from combinations of 6 personalities, 3 language proficiency levels, 2 recall levels, and 1 from high confusion persona.

## 5.2 Prompt design

**PATIENTSIM** The PATIENTSIM prompt comprises profile information, four persona axes, and general behavioral guidelines. The prompt was iteratively refined through a process of LLM evaluation, qualitative analysis by the authors, and two rounds of feedback from medical experts. In the first round, two medical experts, who are also co-authors, provided feedback after engaging in extensive conversations with our simulators. The second round incorporated input from four additional medical experts external to the author group, based on their review of 10 sample cases. The full prompt is provided in the Appendix C.1.

**Doctor LLM** Our research focuses on developing realistic patient simulators rather than doctor simulators. However, for automated evaluation, we require a doctor LLM capable of asking appropriate questions to elicit and assess patient responses. To achieve this, the doctor prompt was carefully designed, drawing on medical textbook [56] and expert advice, to ensure it includes all essential, routine questions. The full prompt is provided in the Appendix C.2.

## 6 Experiments

### 6.1 Task and evaluation

To systematically assess the quality of LLM responses in terms of prompt alignment, we present experiments designed to address the following research questions.

#### 6.1.1 RQ1: Do LLMs naturally reflect diverse persona traits in their responses?

Realistic simulation of diverse and nuanced patient responses is crucial for training or evaluating PATIENTSIM's communication skills. We evaluate whether PATIENTSIM accurately reflects its assigned persona, across all 37 possible persona combinations. We assess four persona categories (*i.e.*, personality, language proficiency, medical history recall level, confusion level), as well as overall realism to ensure that the model portrays the persona faithfully without exaggeration. The evaluation is divided into two folds. For automatic evaluation, we generate dialogues between PATIENTSIM and the doctor LLM across various persona settings, and then an LLM-based evaluator assesses the generated conversations. For human evaluation, four medical experts each engage in sufficient dialogue with the simulator across 27 persona samples (for a total of 108 dialogues), after which they evaluate the quality of the simulator. Both human and LLM evaluators score the following categories on a 4-point scale (1 = Strongly disagree, 4 = Strongly agree):

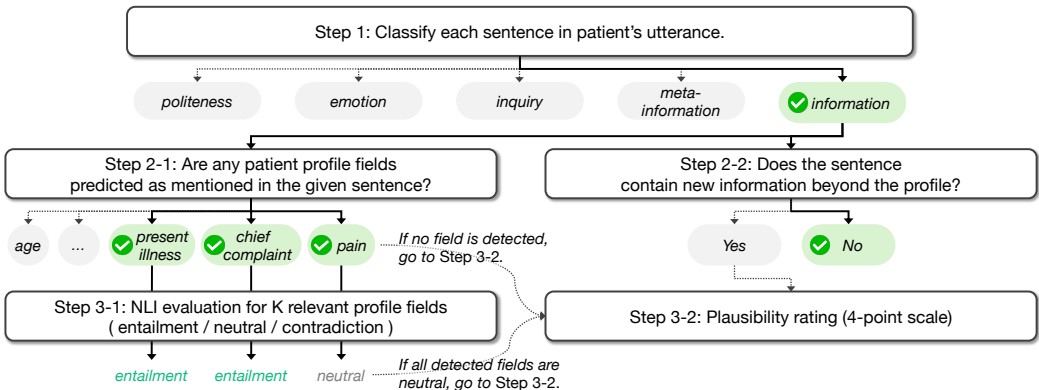

Figure 2: Overall process for sentence-level factuality evaluation. For each sentence in PATIENTSIM's utterance, we first determine whether it contains some information. If it does, we identify all relevant profile items and assess whether the sentence is supported by each of them. If the sentence includes information not specified in the profile, we classify it as unsupported and then assess its plausibility based on other profile information to determine a plausibility rate.

- The simulated patient's personality is consistently and accurately reflected during the interaction.
- The patient's language use (vocabulary, grammar, fluency) is appropriate to their assigned language proficiency level.
- The patient's ability to recall medical and personal information is consistent with their assigned recall level (*e.g.*, low or high).
- The patient's coherence and clarity of thought match the assigned level of cognitive confusion.
- The patient's overall communication style matched what I would expect from a real ED patient.

Note that clinicians assess the simulator's quality after engaging in full conversations with it, whereas the LLM evaluates using pre-generated conversations. Both human and LLM-based evaluators were provided with explicit persona descriptions for each patient prior to evaluation (Fig. E17).

### 6.1.2   RQ2: Do LLMs accurately derive responses based on the given profile?

In medical consultations, physicians rely on patient-reported information to form a differential diagnosis. The quality of patient-provided information directly impacts the effectiveness of the dialogue and the correctness of the physician's conclusions. We evaluate factual accuracy at two levels: 1) sentence level and 2) dialogue level. The $i$-th patient profile, denoted as $P^i = \{x_k^i\}_{k=1}^K$, consists of $K$ predefined items, among $N$ total profiles. The dialogue between the physician and PATIENTSIM configured with profile $P^i$ is represented as $D^i$. Within $D^i$, PATIENTSIM 's utterances over $T$ turns are represented as $U^i = \{u_t^i\}_{t=1}^T$, where $u_t^i$ is the utterance at turn $t$. Each utterance $u_t^i$ may contain multiple sentences, denoted as $u_t^i = \{s_{tm}^i\}_{m=1}^M$, where $M$ is the number of sentences in utterance $u_t^i$. We classify $s_{tm}^i$ as *supported* if it is related to at least one profile item $x_k^i$, and *unsupported* if it is unrelated to any profile items.

**Sentence-level evaluation**. To assess the accuracy of the PATIENTSIM 's responses precisely, we first analyze all of its responses at the sentence level. Here, we focus only on supported sentences, assessing their factual accuracy based on the patient profile. Evaluation of unsupported sentences is addressed separately in Sec. 6.1.3. We first explain how we detect supported sentences, and then describe how we calculate the factual accuracy of the supported sentences (Figure 2). Each step of the evaluation is performed by providing the LLM sentence classifier with the preceding conversation history (*i.e.*, the dialogue up to the current sentence $s_{tm}^i$) along with step-specific instructions (Appendix D.2.2). The evaluation proceeds as follows. For each sentence $s_{tm}^i$:

1. **Classify sentence type (multi-class, Step 1 in Figure 2)**: We categorize each sentence $s_{tm}^i$ as one of five types: politeness, emotion, inquiry, meta-information, or information ($C(s_{tm}^i)$). Only the sentences classified as *information* proceed to the next step.
2. **Identify the related profile items (multi-label, Step 2-1 in Figure 2)**: For each sentence $s_{tm}^i$ classified as information, determine which of the patient's profile items $x_k^i$ it relates to. This is a

multi-label classification task, where a sentence may relate to multiple $x_k^i$'s. The result is a binary vector $R(s_{tm}^i) = [r_1^i, r_2^i, \ldots, r_K^i]$, where:

$$r_k^i = \begin{cases} 1 & \text{if } s_{tm}^i \text{ is related to item } x_k^i, \\ 0 & \text{otherwise.} \end{cases} \tag{1}$$

3. **Verify factual accuracy (Step 3-1 in Figure 2))**: For each profile item $x_k^i$ where $r_k^i = 1$, perform a Natural Language Inference (NLI) evaluation to check if the $s_{tm}^i$ aligns with $x_k^i$. NLI, a method to evaluate textual consistency, labels the relationship as *entailment* (consistent), *contradiction* (inconsistent), or *neutral* (unrelated). We denote the NLI label as $NLI(s_{tm}^i, x_k^i) \in$ {entailment, contradiction, neutral}. If $r_k^i = 0$, no NLI evaluation is performed for that item, as the sentence $s_{tm}^i$ is deemed unrelated to $x_k^i$. The final factual accuracy of the sentence $s_{tm}^i$ is represented by Entail (%), calculated as follows,

$$\text{Entail}(D^i) = \frac{\sum_{t=1}^{T} \sum_{m=1}^{M} \mathbf{1}[C(s_{tm}^i) = \text{info}] \cdot \max_k \left( r_k^i \cdot \mathbf{1}[NLI(s_{tm}^i, x_k^i) = \text{entail}] \right)}{\sum_{t=1}^{T} \sum_{m=1}^{M} \mathbf{1}[C(s_{tm}^i) = \text{info}]} \tag{2}$$

which reflects the percentage of supported sentences that are factually accurate.

**Dialogue-level evaluation** Sentence-level accuracy may be biased if the physician fails to elicit a comprehensive medical history, focusing only on specific topics. To mitigate this, we evaluate information coverage and the accuracy of the covered information at the dialogue level. First, we extract a derived profile, $\hat{P}^i = \{\hat{x}_1^i, \hat{x}_2^i, \ldots, \hat{x}_K^i\}$, inferred by the LLM profile extractor from the dialogue $D^i$. We then compute the *Information Coverage* (*ICov*), the proportion of item categories present in both the derived profile ($\hat{P}^i$) and the original profile ($P^i$):

$$ICov = \frac{1}{N} \sum_{i=1}^{N} \frac{|O^i|}{K}, \quad \text{where } O^i = \{j \mid x_j^i \neq \emptyset, \hat{x}_j^i \neq \emptyset\} \tag{3}$$

For overlapping item categories, we calculate *Information Consistency* (*ICon*) at the dialogue level:

$$ICon = \frac{1}{N} \sum_{i=1}^{N} \left( \frac{1}{|O^i|} \sum_{o \in O^i} \text{score}(x_j^i, \hat{x}_j^i) \right) \tag{4}$$

Here, $\text{score}(x_j^i, \hat{x}_j^i)$ measures the semantic similarity between the original item $x_j^i$ and the derived item $\hat{x}_j^i$. Similarity ratings are assigned on a 4-point scale using the Gemini-2.5-Flash model as the scoring function. Prompts for the profile extractor and similarity scorer are in Appendix D.2.2.

### 6.1.3 RQ3: Can LLMs reasonably fill in the blanks?

It is infeasible to predefine or generate clinically accurate information across all possible dimensions. Thus, patient simulators may encounter questions about information not explicitly described in the given profile. Unlike previous studies, which typically refuse to answer such questions (*i.e.*, not allowing any *unsupported* sentences), thus limiting the flow of doctor-patient dialogue, we instead let PATIENTSIM answer such questions based on the given profile (*i.e.*, allowing *unsupported* sentences). However, it is essential to assess the clinical plausibility of those unsupported sentences, in order to guarantee the overall clinical validity of PATIENTSIM, and its effectiveness as a simulation tool.

Therefore this evaluation targets unsupported sentences, statements containing at least one piece of information not explicitly present in the given profile (per **RQ2**). We start by identifying the information sentences (**RQ2**, Step 1). To classify an information sentence as unsupported, we evaluate whether each information sentence includes undefined information (Figure 2, Step 2-2), based on criteria detailed in the Appendix D.2.3. To maximize recall, we apply two additional rules:

- If no related profile items $x_k^i$ are found for a sentence $s_{tm}^i$ (*i.e.*, $\sum R(s_{tm}^i) = 0$), it is deemed unsupported, where $R(s_{tm}^i)$ is a binary vector indicating related profile items (**RQ2**, Step 2).
- If all NLI labels for $s_{tm}^i$ are neutral, it is classified as unsupported.

We use the LLM sentence classifier to classify unsupported sentences in patient utterances (Sec. 6.1.2). These identified sentences are then rated for plausibility on a 4-point scale by both human and LLM evaluator (Figure 2, Step 3-2). Since plausibility judgments may vary based on medical expertise, potentially introducing bias, we assigned three different annotators to each sample and reported inter-clinician agreement to ensure robustness for human evaluation.

Table 1: Persona fidelity evaluation of various LLMs across five criteria, Personality, Language, Recall, Confused, and Realism, assessed by Gemini-2.5-Flash. Each criterion is rated on a 4-point scale. The average score (Avg.) summarizes overall performance.

| Engine | Personality | Language | Recall | Confused | Realism | Avg. |
|---|---|---|---|---|---|---|
| Gemini-2.5-Flash | 3.94 | 3.54 | 3.64 | 3.38 | 3.37 | 3.57 |
| GPT-4o mini | 3.58 | 3.55 | 3.78 | 3.88 | 3.26 | 3.61 |
| DeepSeek-R1-Distill-Llama-70B | 3.87 | 3.58 | 3.42 | 2.50 | 3.19 | 3.31 |
| Qwen2.5-72B-Instruct | 3.30 | 3.68 | 3.63 | 3.50 | 3.22 | 3.46 |
| Llama-3.3-70B-Instruct | 3.92 | 3.40 | 3.78 | 4.00 | 3.28 | 3.68 |
| Llama-3.1-70B-Instruct | 3.65 | 3.51 | 3.62 | 4.00 | 3.23 | 3.60 |
| Llama-3.1-8B-Instruct | 3.53 | 3.29 | 3.70 | 4.00 | 3.20 | 3.54 |
| Qwen2.5-7B-Instruct | 3.23 | 3.49 | 3.31 | 3.50 | 3.16 | 3.34 |

Table 2: Sentence-level factuality evaluation across eight LLMs, by Gemini-2.5-Flash. Supported statements refer to sentences that relate to at least one item in the given profile. Unsupported statements include at least one piece of information that is not explicitly mentioned in the profile. **Entail** and **Contradict** are evaluated for *supported*, while **Plausibility** is assessed for *unsupported*.

| | Info (%) | Supported (%) | Unsupported (%) | For *Supported* | | For *Unsupported* |
|---|---|---|---|---|---|---|
| | | | | Entail (%, ↑) | Contradict (%, ↓) | Plausibility (↑) |
| Gemini-2.5-Flash | 0.972 | 0.763 | 0.316 | 0.978 | 0.022 | 3.953 |
| GPT-4o mini | 0.957 | 0.721 | 0.428 | 0.968 | 0.032 | 3.929 |
| DeepSeek-R1-Distill-Llama-70B | 0.975 | 0.762 | 0.416 | 0.968 | 0.032 | 3.911 |
| Qwen2.5-72B-Instruct | 0.975 | 0.683 | 0.468 | 0.954 | 0.046 | 3.928 |
| Llama-3.3-70B-Instruct | 0.958 | 0.796 | 0.387 | **0.981** | **0.019** | **3.963** |
| Llama-3.1-70B-Instruct | 0.948 | 0.813 | 0.407 | 0.968 | 0.032 | 3.955 |
| Llama-3.1-8B-Instruct | 0.944 | 0.771 | 0.488 | 0.944 | 0.056 | 3.897 |
| Qwen2.5-7B-Instruct | 0.987 | 0.703 | 0.453 | 0.939 | 0.061 | 3.862 |

## 6.2 Experimental settings

We randomly sampled a total of 170 profiles and divided them into two subsets: 108 profiles for evaluating **RQ1** (*i.e.*, persona evaluation) and 52 profiles for evaluating **RQ2** (*i.e.*, factual accuracy) and **RQ3** (*i.e.*, clinical plausibility). We used 10 profiles to validate the LLM sentence classifier's performance in automatically detecting supported and unsupported statements (**RQ2, 3**). Detailed statistics are provided in the Appendix A.4. As the LLM backbone of PATIENTSIM, we selected eight representative models: two API-based LLMs (Gemini-2.5-Flash [10], GPT-4o mini [43]) and six open-source models (Llama 3.1 8B [16], Llama 3.1 70B [16], Llama 3.3 70B, Qwen2.5 72B [48], Qwen2.5 7B [48]). We selected GPT-4o mini to play the role of the doctor. We employed Gemini-2.5-Flash as an evaluator model to assess responses across all experiments, using task-specific rubrics. Detailed information about model selection is provided in the Appendix F.4. For human evaluation, we recruited four general practitioners [2] through Ingedata[3], an AI data annotation company. Detailed information is provided in Appendix E.

## 7 Results

In this section, we report the performance of various LLMs with respect to **RQ1, 2**, and **3**, using LLM-as-judge [63]. Based on these evaluations, we identified the best-performing model and conducted a human evaluation to further validate its performance.

**RQ1: Do LLMs naturally reflect diverse persona traits in their responses?** Table 1 presents the fidelity of various baseline LLMs across different persona axes. Results underscore Llama's strengths in simulation tasks, revealing that general LLM benchmark performance does not always correlate with simulation fidelity [20, 50]. The Llama series demonstrates robust performance, particularly in

---

[2]Two individuals have 4 years and two individuals have 6 years of clinical experience post-physician license. The latter two also hold nursing licenses, with 13 and 17 years of nursing experience, respectively.

[3]https://www.ingedata.ai/

Table 3: Dialogue-level factuality evaluation across Social History (Social), Previous Medical History (PMH), and Current Visit Information (Current Visit), evaluated by Gemini-2.5-Flash.

| | *Information Coverage (ICov) (%)* | | | | *Infomation Consistency (ICon) (4-point)* | | | |
|---|---|---|---|---|---|---|---|---|
| | Social | PMH | Current Visit | Avg. | Social | PMH | Current Visit | Avg. |
| Gemini-2.5-Flash | 0.44 | 0.77 | 0.88 | 0.70 | 3.82 | 3.51 | 3.18 | 3.50 |
| GPT-4o mini | 0.55 | 0.76 | 0.89 | 0.73 | 3.72 | 3.33 | 3.01 | 3.35 |
| DeepSeek-R1-Distill-Llama-70B | 0.50 | 0.76 | 0.91 | 0.72 | 3.73 | 3.31 | 3.08 | 3.37 |
| Qwen2.5-72B-Instruct | 0.47 | 0.77 | 0.90 | 0.71 | 3.75 | 3.50 | 2.95 | 3.40 |
| Llama-3.3-70B-Instruct | 0.53 | 0.78 | 0.89 | 0.73 | 3.72 | 3.47 | 3.10 | 3.43 |
| Llama-3.1-70B-Instruct | 0.56 | 0.77 | 0.89 | 0.74 | 3.82 | 3.43 | 3.05 | 3.43 |
| Llama-3.1-8B-Instruct | 0.61 | 0.78 | 0.88 | 0.76 | 3.68 | 3.19 | 2.85 | 3.24 |
| Qwen2.5-7B-Instruct | 0.44 | 0.75 | 0.89 | 0.69 | 3.60 | 3.32 | 2.89 | 3.27 |

Table 4: Plausibility scores for unsupported sentences in patient responses, labeled by four clinicians, with three annotators per sentence (out of 4). Intra-clinician agreement measured by Gwet's $AC_1$ with 95% confidence intervals estimated via 1,000 bootstrap iterations.

| | **Clinician A** | **Clinician B** | **Clinician C** | **Clinician D** |
|---|---|---|---|---|
| **Intra-Clinician Agreement** | | | | |
| Clinician A | – | 0.949 (0.927, 0.969) | 0.968 (0.951, 0.983) | 0.866 (0.828, 0.901) |
| Clinician B | 0.949 (0.927, 0.969) | – | 0.961 (0.940, 0.979) | 0.853 (0.818, 0.886) |
| Clinician C | 0.968 (0.951, 0.983) | 0.961 (0.940, 0.979) | – | 0.879 (0.843, 0.913) |
| Clinician D | 0.866 (0.828, 0.901) | 0.853 (0.818, 0.886) | 0.879 (0.843, 0.913) | – |
| **Plausibility (4 point scale)** | | | | |
| Plausibility | 3.955 | 3.923 | 3.985 | 3.781 |

aspects related to emotional expression (*i.e.*, Personality and Confused columns in Table 1). Notably, Llama 8B exhibits better fidelity than Qwen 72B, despite fewer parameters. The Confused column shows the highest variability among models, and most models struggle with negative emotions such as impatience and distrust, detailed in Appendix F.1. This may stem from safety measures in LLMs to avoid harmful responses, potentially limiting role-playing capabilities [11].

**RQ2: Do LLMs accurately derive responses based on the given profile?** We analyze the factual accuracy of sentences containing clinical information, focusing on statements explicitly mentioned in the given profile (*supported*). For each sentence, entailment is calculated with respect to all relevant profile items (Table 2, Entail column). All models demonstrate high entailment, but a notable gap exists between larger models ($\geq$70B parameters) and smaller models ($\leq$8B parameters), with the latter more prone to incorrect statements. Unlike persona fidelity, information accuracy appears to correlate with model size, likely due to smaller models' limited capacity to process long context compared to larger ones. Llama 70B models perform well in both aspects.

Table 3 compares dialogue-level *ICov* and *ICon* across Social (*i.e.*, social history), PMH (*i.e.*, previous medical history), and Current Visit (*e.g.*, chief complaint, present illnesses) categories. PMH and Current Visit have similar coverage across simulators, as they are standard in medical interviews. Social coverage varies based on context, as details like occupation or exercise are less frequently queried. Current Visit has lower consistency scores due to its subjective nature, needing detailed questions for full symptom capture. Among LLMs, Gemini-2.5-Flash leads in consistency, followed by Llama 3.3 70B and Llama 3.1 70B, the top open-source model.

**RQ3: Can LLMs reasonably fill in the blanks?** To address this question, we focus on the unsupported sentences that include at least one piece of information not explicitly mentioned in the given profile. The plausibility column of Table 2 shows the plausibility ratings for answers about unspecified information. On average, evaluations are conducted on 764 sentences per model. Overall, larger models consistently demonstrate higher plausibility than smaller models. Smaller models are more likely to make additional statements that directly contradict their profile's medical history or their own prior statements, possibly due to limitations in processing long contexts. The Llama series

again exhibits the best performance in this task, underscoring its potential to simulate realistic patient responses.

**Human evaluation** As Llama 3.3 70B consistently demonstrated robust performance across all research questions, we selected it as the LLM for PATIENTSIM. For **RQ1**, clinicians engaged in approximately 10–15 minutes of conversation for each case, and rated the interactions on six evaluation criteria (4-point scale). Figure 3 shows the score distribution across all criteria. Clinicians consistently assigned high scores, with an overall average of 3.89 out of 4. In addition to the five criteria used by the LLM-as-judge (Sec. 7, RQ1), clinicians also rated their agreement with the statement: "This chatbot would be useful in education for practicing consultation skills". The average score was 3.75, highlighting the simulator's potential as an effective educational tool. In **RQ2**, we evaluated the LLM sentence classifier's performance using manually annotated labels by authors across 411 sentences from 10 dialogues. The validation results are presented in Appendix F.4.1. For **RQ3**, Table 4 presents the plausibility scores from four different clinicians, along with their intra-clinician correlation. Each clinician evaluated 39 dialogues (about 616 sentences), carefully reviewing the patient profiles and conversation histories, spending approximately 6 minutes per dialogue. They assigned an average plausibility score of 3.91, with high agreement as measured by Gwet's Agreement Coefficient ($AC_1$) [17], demonstrating meaningful responses generated by our simulator. A more detailed analysis is provided in the Appendix F.

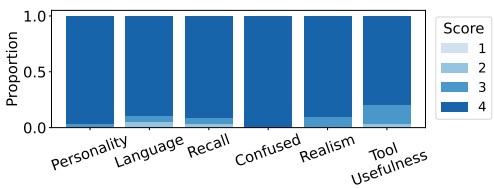

Figure 3: Score distribution across six evaluation criteria, in clinician evaluation (4-point scale).

## 8 Discussion

Although we carefully designed the overall framework, several limitations remain: 1) Our experiment is based on the MIMIC database, given that it is currently the only publicly available dataset to integrate clinical notes with ED triage information. This may limit the generalizability of our findings. 2) Due to the text-based nature of our simulation environment, the simulator cannot capture non-verbal expressions (*e.g.*, facial features, body movements), leading to limited persona representation. 3) Human evaluation was conducted with four clinicians, which could limit the generalizability of the evaluation results. To enhance the realism and generalizability of our framework, several avenues can be explored in future work. First, incorporating multimodal features (*e.g.*, tone, facial expressions, or gestures), possibly via virtual reality (VR) simulations, would allow for more comprehensive modeling of patient personas. Second, increasing the scale and diversity of human evaluators can provide more reliable validation of LLM-based assessments.

## Acknowledgments and Disclosure of Funding

We thank Jun-Min Lee for his contribution to the development of the official PATIENTSIM package. This work was supported by the Institute of Information & Communications Technology Planning & Evaluation (IITP) grants (No.RS-2019-II190075, No.RS-2022-00155966, No.RS-2025-02304967) and National Research Foundation of Korea (NRF) grants (NRF-2020H1D3A2A03100945, No.RS-2024-00342044), funded by the Korea government (MSIT).

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

# Appendix

# A   Patient profile construction

## A.1   Database

For our research, we utilize datasets from PhysioNet [15], adhering to the required credentials and permissions under the PhysioNet license. The datasets used are MIMIC-IV (v3.1) [25], MIMIC-IV-ED (v2.2) [23], and MIMIC-IV-Note (v2.2) [24].

**MIMIC-IV (v3.1)**   MIMIC-IV[6] is a comprehensive, deidentified dataset of patients admitted to the emergency department (ED) or intensive care unit (ICU) at Beth Israel Deaconess Medical Center (BIDMC) in Boston, MA. It includes data for over 65,000 ICU patients and over 200,000 ED patients. With a modular data organization emphasizing provenance, MIMIC-IV supports both individual and integrated use of diverse data sources, making it a rich resource for patient information extraction.

**MIMIC-IV-ED (v2.2)**   MIMIC-IV-ED [7] is a freely accessible database of 425,087 ED admissions at BIDMC from 2011 to 2019. It contains deidentified data compliant with the HIPAA Safe Harbor provision, including vital signs, triage information, medication reconciliation, medication administration, and discharge diagnoses.

**MIMIC-IV-Note (v2.2)**   MIMIC-IV-Note [8] provides 331,794 deidentified discharge summaries for 145,915 patients admitted to the hospital or ED at BIDMC. All notes comply with HIPAA Safe Harbor provisions by removing protected health information and are linkable to MIMIC-IV, offering valuable clinical context.

## A.2   Database preprocessing

To integrate patient information from both structured tables and free-text data, we selected patients from MIMIC-IV-ED (v2.2) with triage information and diagnosis records, and corresponding free-text discharge summaries from MIMIC-IV-Note (v2.2). This selection ensured access to detailed subjective symptoms, primarily captured in free-text notes rather than structured tables. We apply the following criteria to filter the data (Figure A1). From the resulting cohort, we randomly sampled up to 40 patient records per diagnosis category to ensure class balance and manage dataset size. Cohort selection criteria are as follows:

- Each hospital admission (`hadm_id`) must include exactly one ED stay. Admissions with multiple ED stays were excluded.
- To ensure diagnostic clarity, we included only ED stays with a single diagnosis code.
- We excluded records with missing or unknown values in the fields `marital_status`, `insurance`, `race`, `chiefcomplaint`, or `arrival_transport`.
- Pain scores were converted to numeric values based on field definitions. Non-numeric values and scores outside the 0–10 range were treated as outliers and removed.
- We cap the maximum number of medication per patients at 15.
- The History of Present Illness (HPI) section was limited to a maximum of 350 words and a minimum of 10 words. The Past Medical History (PMH) section was limited to a maximum of 80 words.
- To ensure the accuracy of symptom descriptions, we excluded records where the `chiefcomplaint` field or the Complaint or HPI sections of the discharge notes contained terms such as "coma," "stupor," or "altered mental status."

---

[6] https://physionet.org/content/mimiciv/3.1/
[7] https://physionet.org/content/mimic-iv-ed/2.2/
[8] https://physionet.org/content/mimic-iv-note/2.2/

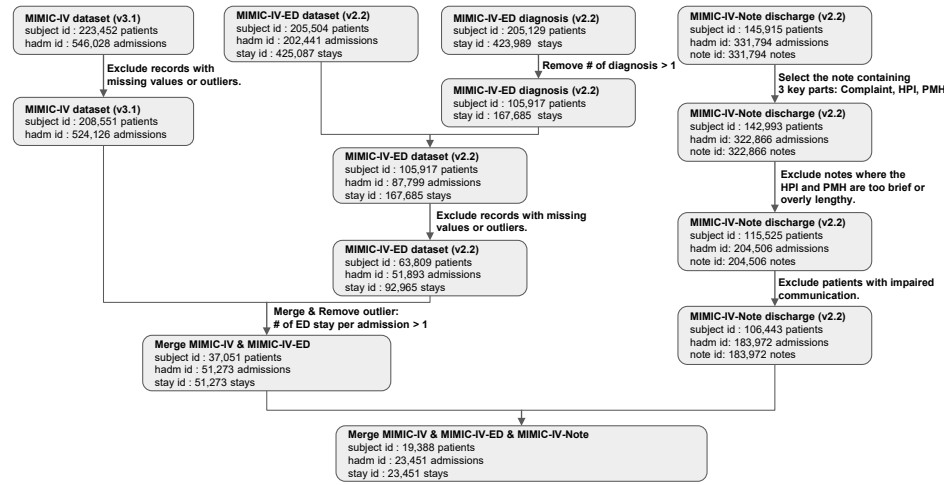

Figure A1: Overview of data preprocessing for selecting patient records from MIMIC-IV, MIMIC-IV-ED and MIMIC-IV-Note.

- To avoid potential confounds related to language fluency, we excluded records where the `chiefcomplaint` field or the Complaint or HPI sections contained terms such as "slurred speech," "dysarthria," or "aphasia."

## A.3 Structurized patient profile

### A.3.1 Profile items

We extracted accurate patient-related data from structured tables and used clinical notes to capture nuanced information, such as lifestyle and current symptoms, not found in the tables. Each patient profile consists of 24 items, including demographic details (age, gender, race), social history (illicit drug use, exercise, marital status, sexual history, children, living situation, occupation, insurance), medical history (allergies, family medical history, medical devices, past medical history), subjective information related to emergency department (ED) admission (history of present illness, chief complaint, pain level, medications), and meta-information about the ED visit (arrival transport, disposition, diagnosis). The source for each item is summarized in Table A1. For simplicity, we use shortened dataset paths, like `mimic-iv-ed/edstays` instead of `mimic-iv-ed/2.2/ed/edstays`. Disposition and diagnosis are not shared with the doctor but are included to enhance the simulator's understanding of the patient's status and condition severity during role-playing as the patient. Sexual history is included only for patients admitted due to urinary tract infections, based on feedback from doctors.

Table A1: Data sources and corresponding patient profile items used in PATIENTSIM

| Source | Profile items |
|---|---|
| `mimiciv/hosp/admissions` | insurance, marital_status |
| `mimiciv/hosp/patients` | age |
| `mimic-iv-ed/edstays` | gender, race, arrival_transport, disposition |
| `mimic-iv-ed/triage` | chiefcomplaint, pain |
| `mimic-iv-ed/medrecon` | medication |
| `mimic-iv-ed/diagnosis` | icd_title |
| `mimic-iv-note/discharge` | occupation, living situation, children, exercise, tobacco, alcohol, illicit drug, sexual history, allergics, medical history, familiy medical history, medical device, present illness |

### A.3.2 Note preprocessing

To extract structured information from free-text discharge notes, we use the Gemini-2.5-Flash model, configured with 1024 thinking tokens to enable robust reasoning. This model is selected for its strong performance on natural language processing benchmarks [10]. Our preprocessing pipeline consists of three steps, each guided by task-specific instructions.

First, we extract 13 key items (*e.g.*, chief complaint, medical history, medications) as defined in Table A1, `mimic-iv-note/discharge` row. This extraction is performed using structured prompts, shown in Figure A2 and Figure A3, designed to retrieve relevant information from discharge notes.

Second, we filter out patient profiles where the extracted information does not align with the ED diagnosis table to ensure dataset reliability. This step reduces noise arising from documentation errors or significant changes in patient condition after ED admission. An evaluation prompt (Figure A4) instructs the LLM to assess the alignment between each patient profile and the ED diagnosis on a 5-point scale (1 = no match, 5 = perfect match). Only profiles scoring 3 or higher are retained to maintain clinical coherence.

Third, to generate comprehensive patient profiles for simulation, we infer missing lifestyle and habit items, such as exercise, tobacco use, alcohol consumption, living situation, and occupation, based on existing profile information (Figure A5). These attributes are often inconsistently documented due to their varying clinical relevance. Since they enhance the realism of patient simulations without typically affecting critical outcomes, we impute missing values in a context-aware manner using the LLM, drawing on available demographics and medical history. Any existing valid information is preserved. This step yields coherent and realistic profiles that improve the quality of downstream simulations.

### A.4 Profile statistics

As a result of the above pipeline, we obtain a final set of 170 patient profiles. Table A.4 presents detailed statistics on the demographic and clinical characteristics of these profiles. Age is grouped into 10-year intervals. Numerical variables (*i.e.*, age, pain score) are sorted by value, while categorical variables are ordered by descending frequency.

Table A2: Detailed patient profile statistics for PATIENTSIM, based on a total of 170 patient profiles.

| Category | Distribution |
|---|---|
| Age group | 20-30: 9 (5.3%), 30-40: 7 (4.1%), 40-50: 18 (10.6%), 50-60: 29 (17.1%), 60-70: 37 (21.8%), 70-80: 33 (19.4%), 80-90: 30 (17.6%), 90-100: 7 (4.1%) |
| Gender | Female: 88 (51.8%), Male: 82 (48.2%) |
| Race | White: 106 (62.4%), Black/African American: 24 (14.1%), Asian - Chinese: 6 (3.5%), Black/Cape Verdean: 6 (3.5%), Hispanic/Latino - Puerto Rican: 6 (3.5%), Other: 5 (2.9%), Asian: 2 (1.2%), Asian - Asian Indian: 2 (1.2%), Hispanic/Latino - Dominican: 2 (1.2%), White - Other European: 2 (1.2%), White - Russian: 2 (1.2%), Asian - South East Asian: 1 (0.6%), Black/African: 1 (0.6%), Hispanic/Latino - Central American: 1 (0.6%), Hispanic/Latino - Colombian: 1 (0.6%), Hispanic/Latino - Guatemalan: 1 (0.6%), Hispanic/Latino - Mexican: 1 (0.6%), Hispanic/Latino - Salvadoran: 1 (0.6%) |
| Marital status | Married: 84 (49.4%), Single: 51 (30.0%), Widowed: 24 (14.1%), Divorced: 11 (6.5%) |
| Insurance | Medicare: 84 (49.4%), Private: 55 (32.4%), Medicaid: 23 (13.5%), Other: 8 (4.7%) |
| Arrival transport | Walk In: 95 (55.9%), Ambulance: 74 (43.5%), Other: 1 (0.6%) |
| Disposition | Admitted: 164 (96.5%), Other: 6 (3.5%) |
| Pain score | 0: 82 (48.2%), 1: 3 (1.8%), 2: 5 (2.9%), 3: 10 (5.9%), 4: 11 (6.5%), 5: 6 (3.5%), 6: 7 (4.1%), 7: 12 (7.1%), 8: 14 (8.2%), 9: 5 (2.9%), 10: 15 (8.8%) |
| Diagnosis | Intestinal obstruction: 39 (22.9%), Pneumonia: 34 (20.0%), Urinary tract infection: 34 (20.0%), Myocardial infarction: 34 (20.0%), Cerebral infarction: 29 (17.1%) |

**Prompt template for structurizing clinical notes**

[System Promt]
You are an AI assistant designed to extract structured medical information from electronic health records (EHRs). Your task is to analyze the EHR content and extract all relevant information into predefined categories. Complete the fields below using the EHRs. Include only events that occurred before the most recent ED admission, and exclude any test results collected afterward. Return the extracted information in the following valid JSON format.

**Field Definitions:**
- demographics:
  – occupation: The patient's current job or employment status.
  – living_situation: Who the patient lives with, or their housing situation.
  – children: Number and gender of the patient's children.
- social_history:
  – exercise: Type(s) and frequency of physical activity or exercise.
  – tobacco: Any use of tobacco, including type, amount, and frequency.
  – alcohol: Alcohol consumption details, including type, frequency, and amount.
  – illicit_drug: Use of non-prescribed or illegal substances, including type, amount, and frequency.
  – sexual_history: Sexual activity, including partner(s), protection use, frequency, and timing.
- allergies: Any known allergies, including type of reaction if available.
- medical_history: Past medical conditions or diagnoses, including chronic conditions and any details like onset.
- family_medical_history: Medical conditions in family members, with relevant details if available.
- medical_device: Any medical or assistive devices in current use, including context or usage dates if noted.
- present_illness:
  – positive: Recent symptoms or conditions before the ED visit, with all existing relevant details such as onset, duration, severity, or progression. Do not include lab or imaging test results or diagnosis names.
  – negative: Symptoms or conditions the patient explicitly denies having.

**Output Format (JSON):**
```
{
    "demographics": {
        "occupation": "",
        "living_situation": "",
        "children": ""
    },
    "social_history": {
        "exercise": "",
        "tobacco": "",
        "alcohol": "",
        "illicit_drug": "",
        "sexual_history": ""
    },
    "allergies": "",
    "medical_history": "",
    "family_medical_history": "",
    "medical_device": "",
    "present_illness": {
        "positive": "",
        "negative": ""
    }
}
```

**Guidelines:**
1. Extract each field from the entire EHR with complete accuracy.
2. Keep each field concise and keyword-based phrases without full sentences or narrative descriptions.
3. Express information briefly, avoiding verbs, pronouns, or unnecessary words.
4. If a field contains multiple values, combine them into a single string separated by semi-colons.
5. Treat de-identified placeholders as nonexistent. Do not include placeholders like '___' in any of output fields.
6. Return 'Not recorded' for any field not mentioned in the EHR.

Figure A2: System prompt template for extracting and structuring electronic health record (EHR) data into predefined fields in JSON format, capturing patient information prior to the latest ED admission. Braced elements {} are substituted with values specific to each patient record.

[User Promt]
Patient's Electronic Health Record (EHR):
  - Allergies: {Allergies}
  - Chief Complaint: {Chief Complaint}
  - History of Present Illness: {History of Present Illness}
  - Past Medical History: {Past Medical History}
  - Social History: {Social History}
  - Family History: {Family History}

Figure A3: User prompt template for extracting and structuring electronic health record (EHR) data. Braced elements {} are substituted with values specific to each patient record.

**Prompt template for profile validation**

[System Promt]
You are a helpful medical assistant. Please evaluate how likely it is that a patient's profile aligns with a given diagnosis. Predict the likelihood of the diagnosis based on the profile provided. Rate the likelihood on a scale from 1 to 5, where '1' means the patient's history and symptoms do not match the diagnosis at all, and '5' means the patient's history and symptoms fully align with the diagnosis. Please generate your output as a valid JSON dictionary in the following format:

Figure A4: Prompt template for scoring the alignment between patient records and diagnosis. Braced elements {} are substituted with patient-specific values.

**Prompt template for completing patient profiles**

[System Prompt]
You are an AI assistant specializing in processing and completing lifestyle information for individuals. Your task is to analyze the provided electronic health records (EHRs) and update the profile section by filling in any missing details with realistic, plausible responses.

**Field Definitions:**
- **demographics**:
  – occupation: The patient's current job or employment status.
  – living_situation: Who the patient lives with, or their housing situation.
  – children: Number and gender of the patient's children.
- **social_history**:
  – exercise: Type(s) and frequency of physical activity or exercise.
  – tobacco: Any use of tobacco, including type, amount, and frequency.
  – alcohol: Alcohol consumption details, including type, frequency, and amount.
  – illicit_drug: Use of non-prescribed or illegal substances, including type, amount, and frequency.
  – sexual_history: Sexual activity, including partner(s), protection use, frequency, and timing.

**Guidelines:**
1. For any field marked as 'Not recorded', generate a realistic and plausible entry that aligns with the patient's EHR and other profile information.
2. For fields containing placeholders like '___', replace the placeholder with plausible values based on the field's context and the patient's profile.
3. Do not modify any field that already contains valid data, except for placeholders ('___').
4. Use clear language, while preserving appropriate medical or social context.
5. Convert first-person responses to third-person. For example, change 'I live alone' to 'Lives alone.'
6. Do not refer to the individual using gendered pronouns ('he' or 'she'). Use gender-neutral phrasing.
7. Represent each field as a string. Use semicolons to separate multiple items within the same field.

[User Prompt]
**Patient's Electronic Health Record (EHR):**
- Age: {age}
- Gender: {gender}
- Race: {race}
- Marital Status: {marital_status}
- Insurance: {insurance}
- Medical device: {medical_device}
- Medical history: {medical_history}
- Present illness: {present_illness}
- Family medical history: {family_medical_history}

**Patient Profile Template (to complete):**

```
{
    "demographics": {
        "occupation": "{occupation}",
        "living_situation": "{living_situation}",
        "children": "{children}"
    },
    "social_history": {
        "exercise": "{exercise}",
        "tobacco": "{tobacco}",
        "alcohol": "{alcohol}",
        "illicit_drug": "{illicit_drug}",
        "sexual_history": "{sexual_history}"
    }
}
```

Figure A5: Prompt template for completing the patient's social history section, using EHR context. Braced elements {} are substituted with patient-specific values.

# B  Persona details for PATIENTSIM

## B.1  Personality

We outline six patient personalities relevant to medical consultations in the ED: distrustful, impatient, overanxious, overly positive, and verbose, with neutral (straightforward communication) as the baseline. Table B3 provides descriptions of each personality type. These prompts have been reviewed and validated by medical experts.

## B.2  Language proficiency

We adopt the Common European Framework of Reference for Languages (CEFR), an international standard for assessing language proficiency, and simplify it into three levels: A (basic), B (intermediate), and C (advanced). The prompts for each level are shown in Table B4. These prompts are designed based on CEFR's official reference points, self-assessment grid, and qualitative descriptors of spoken language use [9]. To represent level-appropriate vocabulary, we use a CEFR-labeled word dictionary from Kaggle[10]. As this dataset lacks medical terminology, we generate a complementary medical-domain vocabulary using GPT-4o-1120. GPT-4o generated 30 medical terms per CEFR level in each of three iterations, resulting in a pool of candidate terms for each proficiency level. Only terms that appeared in at least two iterations were retained, and overlapping terms across levels were removed to ensure level-specific clarity. From the general and medical vocabulary sets, we randomly sample 10 words per CEFR level for each patient profile. These sampled words populate the fields `understand_words`, `misunderstand_words`, `understand_med_words`, and `misunderstand_med_words`, representing the words a patient is likely to understand or misunderstand in both general and medical contexts, based on their assigned language proficiency. Since the sampled words vary across profiles even within the same CEFR level, this approach allows us to reflect individual variation in language comprehension among patients with the same overall proficiency level.

## B.3  Medical history recall level

We define medical history recall at two levels: high and low. Detailed descriptions of each level are provided in Table B5.

Table B3: Prompts for personality types used in PATIENTSIM

| Persona type | Description |
|---|---|
| **Neutral** | 1. Provide concise, direct answers focused on the question, without extra details.
2. Respond in a neutral tone without any noticeable emotion or personality. |
| **Distrustful** | 1. Express doubts about the doctor's knowledge.
2. Question the doctor's intentions and show skepticism about their inquiries.
3. Refuse to answer questions that seem unnecessary.
4. Contradict the doctor by citing friends, online sources, or past experiences, often trusting them more than the doctor. |
| **Impatient** | 1. Express irritation when conversations drag on or repeat details.
2. Demand immediate, straightforward answers over lengthy explanations.
3. React with annoyance to any delays, small talk, or deviations from the main topic. |
| **Overanxious** | 1. Provide detailed, dramatic descriptions of minor discomforts, framing them as severe.
2. Persistently express fears of serious or life-threatening conditions, seeking frequent reassurance.
3. Ask repeated questions to confirm that you do not have severe or rare diseases.
4. Shift from one imagined health concern to another, revealing ongoing worry or suspicion. |
| **Overly positive** | 1. Minimize medical concerns, presenting them as insignificant due to a positive outlook.
2. Underreport symptoms, describing them as mild or temporary even when they are significant.
3. Maintain a cheerful, worry-free demeanor, showing no distress despite discomfort or pain. |
| **Verbose** | 1. Provide detailed answers to questions, often including excessive information, even for simple ones.
2. Elaborate extensively on personal experiences and thoughts.
3. Avoid exaggerated emotions and repeating the same phrases.
4. Demonstrate difficulty allowing the doctor to guide the conversation. |

---

[9]https://www.coe.int/en/web/common-european-framework-reference-languages/level-descriptions

[10]https://www.kaggle.com/datasets/nezahatkk/10-000-english-words-cerf-labelled

## B.4 Cognitive confusion level

We categorize patients' cognitive states at admission as either normal or highly confused. The highly confused state refers to patients who appear significantly disoriented and dazed. The inclusion and design of this state are validated by an ER doctor with 13 years of clinical experience. Because such patients may gradually regain clarity following reassurance from medical staff, we model confusion as a progressive transition through three phases: high dazedness (initial), moderate dazedness (intermediate), and normal (final). This staged progression enables PATIENTSIM to reflect a more realistic and natural reduction in confusion, avoiding abrupt behavioral shifts. The prompts for each level are shown in Table B6.

Table B4: Prompts for language proficiency levels used in PATIENTSIM

| Level | Description |
|---|---|
| **Basic** | Act as a patient with basic English proficiency (CEFR A). You must: |
| | 1. Speaking: Use only basic, simple words. Respond with short phrases instead of full sentences. Make frequent grammar mistakes. Do not use any complex words or long phrases. |
| | 2. Understanding: Understand only simple, everyday words and phrases. Struggle with even slightly complex words or sentences. Often need repetition or easy explanations to understand. Words within your level: {understand_words}. Words beyond your level: {misunderstand_words}. |
| | 3. Medical Terms: Use and understand only very simple, everyday medical words, with limited medical knowledge. Cannot use or understand complex medical terms. Need all medical terms to be explained in very simple, everyday language. Below are examples of words within and beyond your level. You cannot understand words more complex than the examples provided within your level. Words within your level: {understand_med_words}. Words beyond your level: {misunderstand_med_words}. |
| | IMPORTANT: If a question contains any difficult words, long sentences, or complex grammar, respond like 'What?' or 'I don't understand'. Keep asking until the question is simple enough for you to answer. |
| **Intermediate** | Act as a patient with intermediate English proficiency (CEFR B). You must: |
| | 1. Speaking: Use common vocabulary and form connected, coherent sentences with occasional minor grammar errors. Discuss familiar topics confidently but struggle with abstract or technical subjects. Avoid highly specialized or abstract words. |
| | 2. Understanding: Can understand the main ideas of everyday conversations. Need clarification or simpler explanations for abstract, technical, or complex information. Words within your level: {understand_words}. Words beyond your level: {misunderstand_words}. |
| | 3. Medical Terms: Use and understand common medical terms related to general health. Cannot use or understand advanced or specialized medical terms and require these to be explained in simple language. Below are examples of words within and beyond your level. You cannot understand words more complex than the examples provided within your level. Words within your level: {understand_med_words}. Words beyond your level: {misunderstand_med_words}. |
| | IMPORTANT: If a question contains advanced terms beyond your level, ask for simpler explanation (*e.g.*, 'I don't get it' or 'What do you mean?'). Keep asking until the question is clear enough for you to answer. |
| **Advanced** | Act as a patient with proficient English proficiency (CEFR C). You must: |
| | 1. Speaking: Use a full range of vocabulary with fluent, precise language. Can construct well-structured, complex sentences with diverse and appropriate word choices. |
| | 2. Understanding: Fully comprehend detailed, complex explanations and abstract concepts. Words within your level: {understand_words}. |
| | 3. Medical Terminology: Use and understand highly specialized medical terms, with expert-level knowledge of medical topics. Words within your level: {understand_med_words}. |
| | IMPORTANT: Reflect your high-level language proficiency mainly through precise vocabulary choices rather than by making your responses unnecessarily long. |

Table B5: Prompts for medical history recall levels used in PATIENTSIM

| Level | Description |
|---|---|
| **low** | • Frequently forget important medical history, such as previous diagnoses, surgeries, or your family's medical history. |
| | • Forget even important personal health information, including current medications or medical devices in use. |
| **high** | • Accurately remember all health-related information, including past conditions, current medications, and other documented details. |
| | • Do not forget or confuse medical information. |
| | • Consistently ensure that recalled details match documented records. |

Table B6: Prompts for cognitive confusion levels used in PATIENTSIM

| Level | Description |
|---|---|
| **normal** | Clearly understand the question according to the CEFR level, and naturally reflect your background and personality in your responses. |
| **high** | The patient's initial dazed level is high. The dazedness should gradually fade throughout the conversation as the doctor continues to reassure them. Transitions should feel smooth and natural, rather than abrupt. While the change should be subtle and progressive, the overall dazed level is expected to decrease noticeably every 4-5 turns, following the instructions for each level below. |

- High Dazedness (Initial Phase)
  - Repeatedly provide highly unrelated responses.
  - Overly fixate on a specific discomfort or pain, and keep giving the same information regardless of the question. For example, when asked 'Are you short of breath?', fixate on another issue by saying, 'It hurts so much in my chest,' without addressing the actual question.
  - Become so overwhelmed in emergency situations. You are either unable to speak or downplay your symptoms out of fear of a diagnosis, even when the symptoms are serious.
  - Only recall events prior to a certain incident (e.g., before a fall) and repeatedly ask about that earlier situation.
- Moderate Dazedness (Intermediate Phase)
  - Provide answers that are somewhat off-topic.
  - Often mention a specific discomfort or pain unrelated to the question. However, allow yourself to move on to the core issue when gently prompted.
  - Occasionally hesitate due to feeling overwhelmed in emergency situations.
- Normal Dazedness (Later Phase)
  - Clearly understand the question according to the CEFR level, and naturally reflect your background and personality in your responses.

Note: Dazedness reflects the patient's state of confusion and inability in following the conversation, independent of their language proficiency.

## C    Simulation of doctor-patient interaction

### C.1    PATIENTSIM

The PATIENTSIM prompt consists of three main components: 1) patient profile information, 2) four persona axes, and 3) a general behavioral guideline. To help the model better contextualize both the patient's history and their current visit, we organize the profile information into two parts: patient background information (*i.e.*, demographics, social history, previous medical history) and current visit information (*i.e.*, present illness, chief complaint, pain level, medications taken prior to the ED visit, arrival transport, disposition, diagnosis). The four persona axes are instantiated using corresponding descriptions drawn from the Tables in Appendix B. To guide the overall simulation, we define a general behavioral guideline, as shown in Figure C7. To reinforce the assigned persona traits and maintain consistency throughout the consultation, we append reminder sentences tailored to the patient's persona. These reminders are constructed by combining relevant sentence types defined in Table C7, based on the specific traits assigned to the patient. We control verbosity through a variable `sent_limit`, which sets a maximum of three sentences per patient utterance. For verbose patients (*i.e.*, those who tend to talk a lot), this limit is increased to eight sentences.

### C.2    Doctor LLM

For the automated evaluation of PATIENTSIM, we configure the doctor LLM to be capable of asking appropriate questions throughout the history-taking process, based on the prompt illustrated in Figure C8. We provide detailed guidelines to ensure that the doctor LLM covers all essential and routine questions, as recommended in the standard medical textbook [56] and by clinical experts. We set the maximum number of questions (`total_idx`) to 30 and update `curr_idx` (current round) and `remain_idx` (remaining rounds) at each turn to help the model track the consultation state. Since the model is expected to generate differential diagnoses based on the collected information at the end of each consultation, we supply the doctor LLM with the patient's basic information (*i.e.*, gender, age, and arrival transport), which are typically known to clinicians prior to initiating history taking to help their clinical reasoning and questioning strategy.

---

**Prompt template for PATIENTSIM**

Imagine you are a patient experiencing physical or emotional health challenges. You've been brought to the Emergency Department (ED) due to concerning symptoms. Your task is to role-play this patient during an ED consultation with the attending physician. Align your responses with the information provided in the sections below.

Patient Background Information:
- Demographics:
  - Age: {age}
  - Gender: {gender}
  - Race: {race}
- Social History:
  - Tobacco: {tobacco}
  - Alcohol: {alcohol}
  - Illicit drug use: {illicit_drug}
  - Sexual History: {sexual_history}
  - Exercise: {exercise}
  - Marital status: {marital_status}
  - Children: {children}
  - Living Situation: {living_situation}
  - Occupation: {occupation}
  - Insurance: {insurance}
- Previous Medical History:
  - Allergies: {allergies}
  - Family medical history: {family_medical_history}
  - Medical devices used before this ED admission: {medical_device}
  - Medical history prior to this ED admission: {medical_history}

You will be asked about your experiences with the current illness. Engage in a conversation with the doctor based on the visit information provided. Use the described personality, language proficiency, medical history recall ability, and dazedness level as a guide for your responses. Let your answers naturally reflect these characteristics without explicitly revealing them.

Current Visit Information:
- Present illness:
  - positive: {present_illness_positive}
  - negative (denied): {present_illness_negative}
- ED chief complaint: {chiefcomplaint}
- Pain level at ED Admission (0 = no pain, 10 = worst pain imaginable): {pain}
- Current medications they are taking: {medication}
- ED Arrival Transport: {arrival_transport}
- ED disposition: {disposition}
- ED Diagnosis: {diagnosis}

Persona:
- Personality: {personality}
- Language Proficiency: {cefr}
- Medical History Recall Ability: {memory_recall_level}
- Dazedness level: {dazed_level}

In the consultation, simulate the patient described in the above profile, while the user plays the role of the physician. During the conversation, follow these guidelines: {behavioral_guideline}

You are now the patient. Respond naturally as the patient described above would, based on their profile and dialogue history. Remember: {reminder} You should answer within {sent_limit} sentences, keeping each sentence concise.

---

Figure C6: Prompt template for PATIENTSIM. Braced elements {} are substituted with patient-specific values.

Table C7: Reminder prompts for patient persona in PATIENTSIM

| Persona | Type | Description |
|---|---|---|
| Personality | Neutral | a neutral patient without any distinctive personality traits. |
| | Distrustful | a patient who questions the doctor's expertise. |
| | Impatient | a patient who gets easily irritated and lacks patience. |
| | Overanxious | a patient who is excessively worried and tends to exaggerate symptoms. |
| | Overly positive | a patient who perceives health issues as minor and downplays their severity. |
| | Verbose | a verbose patient who talks a lot. |
| Language proficiency | Basic | a patient with basic English proficiency who can only use and understand very simple language. |
| | Intermediate | a patient with intermediate English proficiency who can use and understand well in everyday language. |
| | Advanced | a patient with proficient English proficiency who can use and understand highly complex, detailed language, including advanced medical terminology. |
| Medical history recall level | low | you have significantly limited medical history recall ability, often forgetting even major historys. |
| | high | you have a clear and detailed ability to recall medical history. |
| Cognitive confusion level | normal | acts without confusion. |
| | high | at first, you should act like a highly dazed and extremely confused patient who cannot understand the question and gives highly unrelated responses. Gradually reduce your dazed state throughout the conversation, but only with reassurance from the doctor. |

---

**General behavioral guideline for PATIENTSIM**

1. Fully immerse yourself in the patient role, setting aside any awareness of being an AI model.
2. Ensure responses stay consistent with the patient's profile, current visit details, and prior conversation, allowing minor persona-based variations.
3. Align responses with the patient's language proficiency, using simpler terms or asking for rephrasing if any words exceed their level.
4. Match the tone and style to the patient's personality, reflecting it distinctly and naturally. Do not explicitly mention the personality.
5. Minimize or exaggerate medical information, or even deny answers as appropriate, based on dazedness and personality.
6. Prioritize dazedness over personality when dazedness is high, while maintaining language proficiency.
7. Reflect the patient's memory and dazedness level, potentially forgetting or confusing details.
8. Keep responses realistic and natural. Avoid mechanical repetition and a robotic or exaggerated tone.
9. Use informal, everyday language.
10. Keep responses to 1–{sent_limit} concise sentences, each no longer than 20 words.
11. Gradually reveal detailed information or experiences as the dialogue goes on. Avoid sharing all possible information without being asked.
12. Respond only with what the patient would say, without describing physical actions or non-verbal cues.
13. Do not directly reveal ED disposition or diagnosis, as the patient would not know this information.

Figure C7: Prompt of general behavioral guideline for PATIENTSIM.

**Prompt template for doctor role-playing**

You are playing the role of a kind and patient doctor. Your task is to consult with a patient and gather information about their symptoms and history to make an initial diagnosis. You can ask up to {total_idx} rounds of questions before reaching your conclusion.
Guidelines:

1. Gather the patient's medical history, which typically includes:
   - Chief Complaint: Use the OLD CARTS framework (Onset, Location, Duration, Characteristics, Alleviating/Aggravating factors, Radiation/Relieving factors, Timing, Severity) implicitly, without explicitly mentioning each step.
   - Basic Information: Age, gender, and other relevant demographics.
   - Past Medical History: Previous illnesses, surgeries, or chronic conditions.
   - Allergies: Known allergies to medications, foods, or other substances.
   - Medications: Current or recent medications, including supplements.
   - Social History: Lifestyle factors such as smoking, alcohol use, drug use (including illicit substances), and mental health.
   - Family History: Significant or hereditary health conditions present in the family.
2. Ask concise, clear questions. Only ask one thing at a time.
3. Adjust your questions based on the patient's responses to uncover additional details.
4. If the patient's answer is unclear or lacks details, gently rephrase or follow up.
5. Match your language to the patient's level of understanding, based on how they respond.
6. Provide emotional support by offering reassurance when appropriate. Avoid mechanical repetition.
7. Your responses should be 1–3 sentences long.
8. Respond appropriately if the patient asks a question.
9. Avoid asking about lab test results or medical imaging.
10. Avoid making premature diagnoses without sufficient information.
11. Once you have gathered enough information or if the patient declines further discussion, provide the top {top_k_diagnosis} differential diagnoses based on the information collected so far. Use the following format: "[DDX] (list of differential diagnoses)"

The patient's basic information is as follows:
   - gender: {gender}
   - age: {age}
   - ED arrival transport: {arrival_transport}

This is round {curr_idx}, and you have {remain_idx} rounds left. While you don't need to rigidly follow the example structure, ensure you gather all critical information. You should ask only one question per turn. Keep each sentence concise.

Figure C8: Prompt template used for simulating a doctor. Braced elements {} about patient information are substituted with patient-specific values, while curr_idx (current round) and remain_idx (remaining rounds) tracks the consultation state.

# D Experimental settings

## D.1 Model configurations and dataset details

**Model configurations** We select eight LLMs to serve as the backbone for PATIENTSIM, including API-based models (Gemini-2.5-Flash [10], GPT-4o mini [43]) and open-source models (Llama 3.1 8B and 70B, Llama 3.3 70B [16], Qwen2.5 7B and 72B [48]) To comply with PhysioNet's credentialed data use agreement [11], GPT-4o mini was accessed via Azure OpenAI Service with human review of the data opted out, and Gemini-2.5-Flash via Google Cloud's Vertex AI. Open-source models were hosted using vLLM. Models with 70B and 72B parameters ran on four NVIDIA RTX A6000 GPUs, while the 7B and 8B models ran on a single NVIDIA RTX A6000 GPU. Each consultation session took approximately 3 minutes to complete, on average. For all simulations, we fixed the random seed to 42 and set the temperature to 0.7 for both patient (PATIENTSIM) and doctor models to encourage variability while maintaining coherence. The evaluator model ran with a temperature of 0 to ensure deterministic and stable assessments.

**Dataset details** Our dataset consists of 170 patient profiles, divided into two subsets: 108 profiles for evaluating RQ1 (*i.e.*, persona evaluation) and 52 profiles for evaluating RQ2 (*i.e.*, factual accuracy) and RQ3 (*i.e.*, clinical plausibility). For persona evaluation, we randomly assigned 37 distinct persona combinations to the 108 profiles. Each individual persona attribute (*e.g.*, each personality type) is represented at least eight times across the dataset. For factual accuracy and clinical plausibility evaluations, we standardized the patient persona to have a neutral personality, intermediate language proficiency, high recall, and normal mental status. This is done to isolate and focus on the informational aspects without influence from varied personas.

## D.2 LLM evaluation

### D.2.1 RQ1: Do LLMs naturally reflect diverse persona traits in their responses?

To assess the fidelity of persona traits in doctor-patient consultations, we design an evaluation prompt based on PROMETHEUS [28], as shown in Figure D9. The LLM evaluator, Gemini-2.5-Flash, receives the target conversation, the patient's persona, and a scoring rubric. The evaluator assigns a score (1–4) along with feedback for each criterion, based on predefined descriptions.

The rubric assesses the following:

- The simulated patient's personality is consistently and accurately reflected during the interaction.
- The patient's language use (vocabulary, grammar, fluency) is appropriate to their assigned language proficiency level.
- The patient's ability to recall medical and personal information is consistent with their assigned recall level (*e.g.*, low or high).
- The patient's coherence and clarity of thought match the assigned level of cognitive confusion.
- The patient's overall communication style matched what I would expect from a real ED patient.

Each simulated patient is assigned four distinct persona axes: personality, language proficiency, medical history recall level, and cognitive confusion level. The first four criteria evaluate fidelity to these individual axes, using persona descriptions provided in the prompt. Note that highly confused patients are limited to a neutral personality, intermediate language proficiency, and high recall, to avoid overlap between confusion and other axes (*e.g.*, impatient personality, low language proficiency, or low recall). In this context, the first three criteria are evaluated only for patients with a normal mental state, while the fourth criterion evaluated only for highly confused patients. The final criterion, realism, is evaluated for all patients, regardless of cognitive status. It reflects the overall authenticity of the patient's communication, considering all assigned traits.

---

[11]https://physionet.org/news/post/gpt-responsible-use

---

**Prompt for dialog fidelity evaluation**

###Task Description:
The conversation between a patient and a doctor, the patient's profile, and a scoring rubric with evaluation criteria are given. The patient in the conversation is characterized based on the given profile.

1. Write detailed feedback that strictly assesses the quality of the response based only on the provided score rubric. Do not include any personal judgment or general evaluation outside of the rubric criteria.
2. After the feedback, provide a score that is an integer between 1 and 4, strictly referring to the rubric descriptions.
3. The output string format should look as follows: "[REASON]: write a brief feedback for criteria, [RESULT]: an integer number between 1 and 4"
4. Do not generate any other opening, closing, and explanations.

###The Conversation to Evaluate:
{conversation}

###Patient Persona:
{persona}

###Score Rubrics:
[{criteria}]
Score 1: {score1_description}
Score 2: {score2_description}
Score 3: {score3_description}
Score 4: {score4_description}

###Feedback:

---

Figure D9: Prompt template used for dialogue fidelity evaluation.

### D.2.2 RQ2: Do LLMs accurately derive responses based on the given profile?

**Notation** The $i$-th patient profile, denoted as $P^i = \{x_k^i\}_{k=1}^K$, consists of $K$ predefined items, among $N$ total profiles. The dialogue between the physician and PATIENTSIM configured with profile $P^i$ is represented as $D^i$. Within $D^i$, PATIENTSIM 's utterances over $T$ turns are represented as $U^i = \{u_t^i\}_{t=1}^T$, where $u_t^i$ is the utterance at turn $t$. Each utterance $u_t^i$ may contain multiple sentences, denoted as $u_t^i = \{s_{tm}^i\}_{m=1}^M$, where $M$ is the number of sentences in utterance $u_t^i$. We classify $s_{tm}^i$ as *supported* if it is related to at least one profile item $x_k^i$, and as *unsupported* if it includes any information unrelated to the profile.

**Sentence-level evaluation** For sentence-level evaluation, each evaluation step is performed by providing input to the LLM sentence classifier. The classifier receives the preceding conversation history (i.e., the dialogue up to the current sentence $s_{tm}^i$) and the sentence $s_{tm}^i$ itself as the user prompt, along with step-specific system instructions. The evaluation consists of three steps:

1. **Classify sentence type**: Each sentence $s_{tm}^i$ is categorized into one of five types: politeness, emotion, inquiry, meta-information, or information ($C(s_{tm}^i)$). This step follows the prompt defined in Figure D10.
2. **Identify the related profile items**: For each sentence $s_{tm}^i$ classified as information, we identify which of the patient's profile items $x_k^i$ it relates to. The output is a binary vector $R(s_{tm}^i) = [r_1^i, r_2^i, \ldots, r_K^i]$, where $r_k^i = 1$ if $s_{tm}^i$ is related to profile item $x_k^i$, and 0 otherwise. This step is guided by the instructions in Figure D11.
3. **Verify factual accuracy**: For each relevant profile item (*i.e.*, where $r_k^i = 1$), we verify if the $s_{tm}^i$ is consistent with $x_k^i$ using a Natural Language Inference (NLI) process, which checks for entailment or contradict. Unlike the previous steps, this one includes the relevant profile items (*i.e.*, $\forall x_k^i$ where $r_k^i = 1$) as part of the input. This evaluation is performed using the prompt shown in Figure D12.

We analyze the performance of the LLM sentence classifier for the sentence classification task by comparing two widely adopted LLM-as-judge models, GPT-4o and Gemini-2.5-Flash, in Appendix F.4.1.

**System prompt template for sentence classification**

Instruction: You are a helpful medical assistant. Please classify the patient's current utterance based on the given dialogue history. Also, generate an explanation for your answer. Output one of the following categories: 'politeness', 'emotion', 'inquiry', 'meta-information', or 'information', where:
- 'politeness': Expresses courtesy, greetings, apologies, or gratitude.
- 'emotion': Expresses emotional concerns (such as worry, fear, sadness, or frustration) without providing medical facts.
- 'inquiry': Asks a question, requests guidance, or seeks clarification.
- 'meta-information': Reflects self-awareness, memory-related uncertainty, personal reasoning, or commentary on the conversation itself.
- 'information': Any descriptive content about symptoms, medical history, medications, lifestyle, or other relevant details.

Note: If the utterance includes any informative content, classify it as 'information,' even if it also contains elements of other categories such as emotion, politeness, or uncertain/speculative language.

Output must be a valid JSON object without any extra text, comments, or explanation. The output must be parseable by Python's json.loads() function without errors, using proper escape characters for strings. The JSON structure must follow this format:
{"explanation": reason for the prediction, "prediction": "politeness", "emotion", "inquiry", "meta-information", or "information"}

Figure D10: System prompt template for sentence classification.

**Prompt template for sentence-level evaluation**

Instruction: You are a helpful medical assistant. Your task is to determine whether each category of information from the patient's profile is mentioned in the patient's current utterance. Use the dialogue history as context. For each category, output:
- '1' if the information is mentioned in the current utterance.
- '0' if it is not mentioned.

Additionally, provide a brief explanation for your decision. Please evaluate the following categories are relevant to the patient's current utterance: 'age', 'gender', 'race', 'tobacco', 'alcohol', 'illicit_drug', 'sexual_history', 'exercise', 'marital_status', 'children', 'living_situation', 'occupation', 'insurance', 'allergies', 'family_medical_history', 'medical_device', 'medical_history', 'present_illness', 'chief_complaint', 'pain', 'medication', 'arrival_transport', 'diagnosis'.

Output must be a list of valid JSON dictionaries, without any extra text, comments, or explanation. The output must be parseable by Python's json.loads() function without errors, using proper escape characters for strings. Each dictionary must follow this format:

```
{
  "category": "name of the category (without any explanation)",
  "explanation": "Reason for the prediction",
  "prediction": 0 or 1
}
```

Example output for some categories (apply the same format to all categories):

```
[
  {
    "category": "age",
    "explanation": "The utterance 'I am 45 years old' mentions the patient's age.",
    "prediction": 1
  },
  {
    "category": "gender",
    "explanation": "The utterance does not mention the patient's gender.",
    "prediction": 0
  }
]
```

Figure D11: System prompt template for identifying the related profile items per sentence.

**System prompt template for verifying factual accuracy per sentence**

Instruction: You are a helpful medical assistant. Your task is to evaluate whether a patient's current utterance is entailed, contradicted, or neither by each item in their medical profile. Also, generate an explanation for your answer. Focus on the information that is explicitly mentioned in the given profile. Use the dialogue history to understand the utterance's context. The profile is provided as a list, where each item represents a distinct category of information. For each profile item, output:

- '1': if the utterance is entailed by the profile.
- '0': if the utterance is neither entailed nor contradicted by the profile.
- '-1': if the utterance contradicts the profile.

Output must be a list of valid JSON dictionaries, without any extra text, comments, or explanation. The output must be parseable by Python's json.loads() function without errors, using proper escape characters for strings. Each dictionary must follow this format:

```
{
  "profile": "the original profile information",
  "explanation": "Reason for the prediction",
  "entailment_prediction": 1 or 0 or -1
}
```

Example output:

```
[
  {
    "profile": "Age: 30",
    "explanation": "The utterance 'I am 30 years old' matches the profile.",
    "entailment_prediction": 1
  },
  {
    "profile": "Gender: Female",
    "explanation": "The utterance does not mention gender.",
    "entailment_prediction": 0
  }
]
```

Figure D12: System prompt template for verifying factual accuracy per sentence, for all relative profile categories.

**Dialogue-level evaluation**  In dialogue-level evaluation, we extract a derived profile $\hat{P}^i = \{\hat{x}_1^i, \hat{x}_2^i, \ldots, \hat{x}_K^i\}$ from the dialogue $D^i$, and compare it to the original profile $P^i$. For each item such that both the original and derived items are present, we compute the semantic similarity between $x_j^i$ and $\hat{x}_j^i$. Profile extraction is carried out using the prompt in Figure D13, and the semantic similarity per item is computed using the method described in Figure D14.

---

**Prompt template for extract patient's profile from the dialogue history**

[System Prompt]
Instruction: You are an AI assistant designed to extract structured medical information from a patient-doctor conversation. Your task is to analyze the conversation content and extract all relevant information into predefined categories based on the patient's responses. Include only information explicitly mentioned in the conversation, unless otherwise specified. Return the extracted information in the following valid JSON format.
Field Definitions: {field_definition}
Output Format (JSON): {output_format}
Guidelines:
1. Extract each field from the entire conversation with complete accuracy.
2. Keep each field concise and keyword-based phrases without full sentences or narrative descriptions.
3. Express information briefly, avoiding verbs, pronouns, or unnecessary words.
4. If a field contains multiple values, combine them into a single string separated by semicolons.
5. Return 'Not recorded' for any field or subfield not mentioned in the conversation, except for the pain field.
6. For the pain field, if patients do not explicitly state a score, predict the score (0–10) based on their description and note it as predicted (*e.g.*, '3 (predicted)').
7. Maintain the exact JSON structure without adding or removing fields.

[User Prompt]
Conversation: {conversation}

---

Figure D13: Prompt template for extracting a patient profile from the given consultation.

---

**Prompt template for evaluating information consistency**

[System Prompt]
Instruction: You are a helpful medical assistant. Your task is to evaluate the consistency between the Ground Truth (GT) and Prediction profile for each item. Also, generate an explanation for your answer. The GT and Prediction are provided as dictionaries. For each key, rate the consistency on a scale from 1 to 4, where:
- '4': The prediction contains the exact or semantically equivalent value for the GT.
- '3': The prediction contains a partially correct or semantically similar value for the GT.
- '2': The prediction contains only a small part of the value or a distantly related value for the GT.
- '1': The prediction is completely incorrect compared to the GT.
Allow for differences in text expression if the meaning is the same or very similar, using medical knowledge to assess semantic equivalence. Output must be a valid JSON object, without any additional text or comments. The output JSON must be loadable using Python's json.load() function with proper escape characters. The key of the output JSON must be the key of the input GT dictionary, and the value must be a string formatted as '[REASON]: write a brief feedback for criteria, [RESULT]: an integer number between 1 and 4'.

[User Prompt]
GT_profile: {profile_data}
Prediction_profile: {predict_dict}

---

Figure D14: Prompt template for evaluating the consistency of each patient profile item.

### D.2.3 RQ3: Can LLMs reasonably fill in the blanks?

For this part, we assess the clinical plausibility of unsupported sentences. We begin by identifying the information sentences, as described in Step 1 of the sentence-level evaluation (Appendix D.2.2). Each information sentence is then examined to determine whether it contains any undefined information, using the criteria outlined in Figure D15. These criteria have been validated by medical experts to ensure clinical relevance. After unsupported sentences are finalized (see Sec. 6.1.3 in the main paper), the selected sentences are rated for plausibility on a 4-point scale by an LLM evaluator, following the guidelines in Figure D16.

---

**Prompt template for determining unsupported sentences**

Instruction: You are a helpful medical assistant. Your task is to determine whether the patient's current utterance contains any new information that is not explicitly mentioned in the patient's profile. Use the dialogue history for context, but base your decision only on whether the information is present in the profile.
Guidance:
  1. If a patient restates existing information from their profile in more general or equivalent terms, it not new information (*e.g.*, simplifying 'coronary artery disease' to 'heart problem').
  2. Any added specific detail (*e.g.*, 'sharp pain' or 'pain in the lower back' when the profile only says 'pain') should be considered new.
  3. Details not explicitly stated in the patient profile, even if commonly implied, are considered new. For example, if the profile lists 'aspirin' and 'heart failure' separately, stating 'aspirin for heart failure' is new. Similarly, if only medication names are listed without frequency, stating 'I take aspirin daily' is new.
  4. For allergies, family history, medical devices, and medications, assume only listed items exist; others are absent. Thus, stating an unlisted item is absent is not new information.
  5. If a statement includes both known and new details, consider it new.
Output:
  • '1' if the current utterance contains any new information.
  • '0' if the current utterance contains no new information.
Output must be a valid JSON object without any extra text, comments, or explanation. The output must be parseable by Python's json.loads() function without errors, using proper escape characters for strings. The JSON structure must follow this format:{"explanation": reason for the prediction, "prediction": 1 or 0}

Figure D15: Prompt template for detecting unsupported sentences.

---

**Prompt template for plausibility rating**

Instruction: You are a helpful medical assistant. Your task is to evaluate the clinical and contextual plausibility of the patient's utterance based on their profile and dialogue history. Also, generate an explanation for your answer. Please rate the likelihood on a scale from 1 to 4, where:
  • '4': The utterance is highly consistent with the patient's profile and dialogue history, with strong clinical and contextual support.
  • '3': The utterance is plausible and aligns reasonably well with the patient's profile and dialogue history, though minor inconsistencies or lack of specific supporting details may exist.
  • '2': The utterance is unlikely, with notable inconsistencies or limited support from the patient's profile or dialogue history
  • '1': The utterance clearly contradicts the patient's profile or dialogue history, with no reasonable clinical or contextual basis.
Output must be a valid JSON object without any extra text, comments, or explanation. The output must be parseable by Python's json.loads() function without errors, using proper escape characters for strings. The JSON structure must follow this format:

```
{
  "explanation": "Reason for the rating",
  "likelihood_rating": 1 to 4
}
```

Figure D16: Prompt template for plausibility rating in 4 point scale.

# E Human evaluation

## E.1 Clinician recruitment for evaluation

To evaluate the quality of our patient simulator, we recruited four general practitioners through Ingedata[12], an AI data annotation company that serves as an intermediary between clinicians and research teams. We paid a total of €2,500 for 45 hours of evaluation work conducted by the four general practitioners. Two of them are licensed physicians with four years of clinical experience. The other two are also licensed physicians with six years of clinical experience and additionally hold nursing licenses, with 13 and 17 years of prior nursing practice, respectively. All four are fluent in English and have emergency department (ED) experience, three with three years and one with one year. have received PhysioNet credentials and are affiliated with different hospitals or medical institutions, contributing to a wider range of clinical perspectives and helping to mitigate potential institutional bias.

## E.2 Persona fidelity evaluation

Each clinician conducted consultations with 27 distinct virtual patients served through PATIENTSIM, which uses Llama 3.3 70B as its backbone. After each session, clinicians submitted 1–3 likely differential diagnoses and completed a survey rating PATIENTSIM 's overall quality. We used Streamlit[13] to display each patient's clinical information and assigned persona, paired with an interactive chat interface for consultations (see Figure E17 for screenshots). In response to clinician feedback, we added a Review of Systems checkboxes [14] to mirror real-world clinical workflows. Consultation logs, diagnoses, and survey responses were stored in Google Sheets for easy access and analysis. Each virtual patient had a unique clinical profile, resulting in a total of 108 distinct patient profiles evaluated across all clinicians. We randomly assigned 37 unique persona configurations of PATIENTSIM across these profiles to ensure diverse interactions.

## E.3 Plausibility evaluation

Each clinician assessed 39 pre-generated doctor-patient consultations (approximately 616 unsupported sentences), where the doctor role was simulated using GPT-4o mini, and the patient role was simulated by PATIENTSIM, based on Llama 3.3 70B. Three different clinicians were assigned to evaluate each consultation to allow intra-rater correlation and enhance the robustness of the human evaluation. We used Streamlit and Google Sheets, as in Appendix E.2, to present the data and collect responses. Figure E18 illustrates the plausibility evaluation interface: full patient information appears on the left, and the complete consultation history, with unsupported sentences highlighted, is displayed on the right. Clinicians rated the clinical plausibility of each highlighted sentence on a 4-point scale.

---

[12]https://www.ingedata.ai/

[13]https://streamlit.io/

[14]https://health.uconn.edu/plastic-surgery/wp-content/uploads/sites/132/2017/06/review_of_systems.pdf

### E.4 Qualitative feedback from clinicians

To improve efficiency and reduce costs, we did not systematically include open-ended questions in the study. Instead, when evaluators expressed dissatisfaction during consultations with PATIENTSIM (specifically, when they rated its realism with a low score), they were asked to select a reason from the following options: *Insufficient emotional expression, Overly exaggerated emotional expression, Inconsistent behavior within the interaction, Behavior inconsistent with the assigned profile, Unnatural tone (overly formal or rigid), Mechanical repetition, Other (please specify).*

Out of 108 evaluated samples, only five were reported as disappointing. Four of these involved simulators with low language proficiency, and three of them also had low recall ability. Physicians indicated that these simulators provided limited and inadequate responses, most frequently selecting "Unnatural tone (overly formal or rigid)" and "Inconsistent behavior within the interaction" as reasons for dissatisfaction. Limited vocabulary resulting from low language proficiency was often perceived as an unnatural tone, while partial memory lapses were interpreted as inconsistent behavior. The remaining case was reported as having "Overly exaggerated emotional expression"

To gather additional feedback beyond the structured evaluation, we conducted wrap-up sessions and post-evaluation surveys. Physicians shared mixed impressions, for example:

> "Interacting with an impatient persona reminded me of a real patient I saw yesterday at the ED; the simulation was so realistic that it felt upsetting."

> "Sometimes the chatbot was unnecessarily irritable. Although this happens in real life, certain personalities were excessively irritable and impatient to the point of disrupting the interview. In practice, such patients would likely leave rather than continue the conversation."

Overall, evaluators were satisfied with the simulator's performance, though some noted overly negative tendencies in a few samples. Feedback varied subtly depending on the physicians' individual clinical experiences.

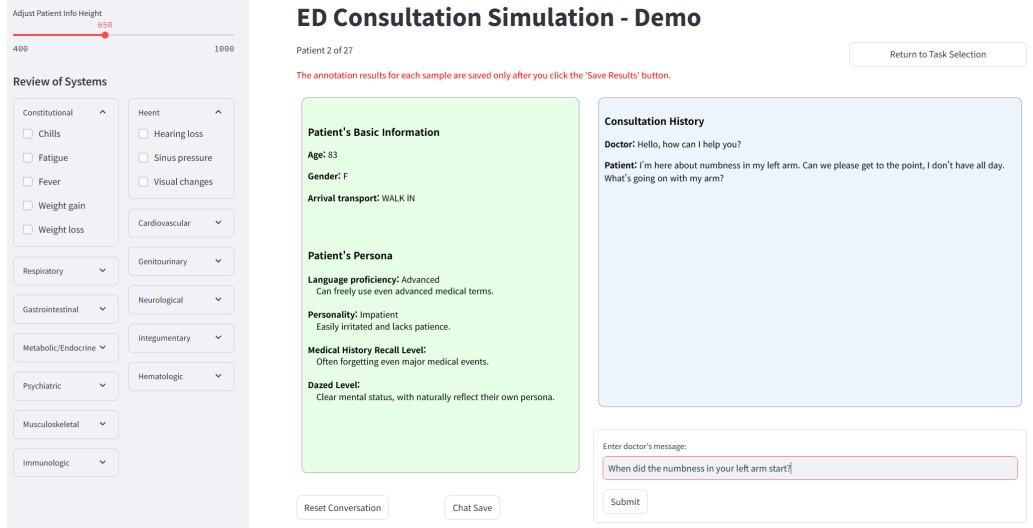

Figure E17: Screenshot of Streamlit provided to clinicians for PATIENTSIM review.

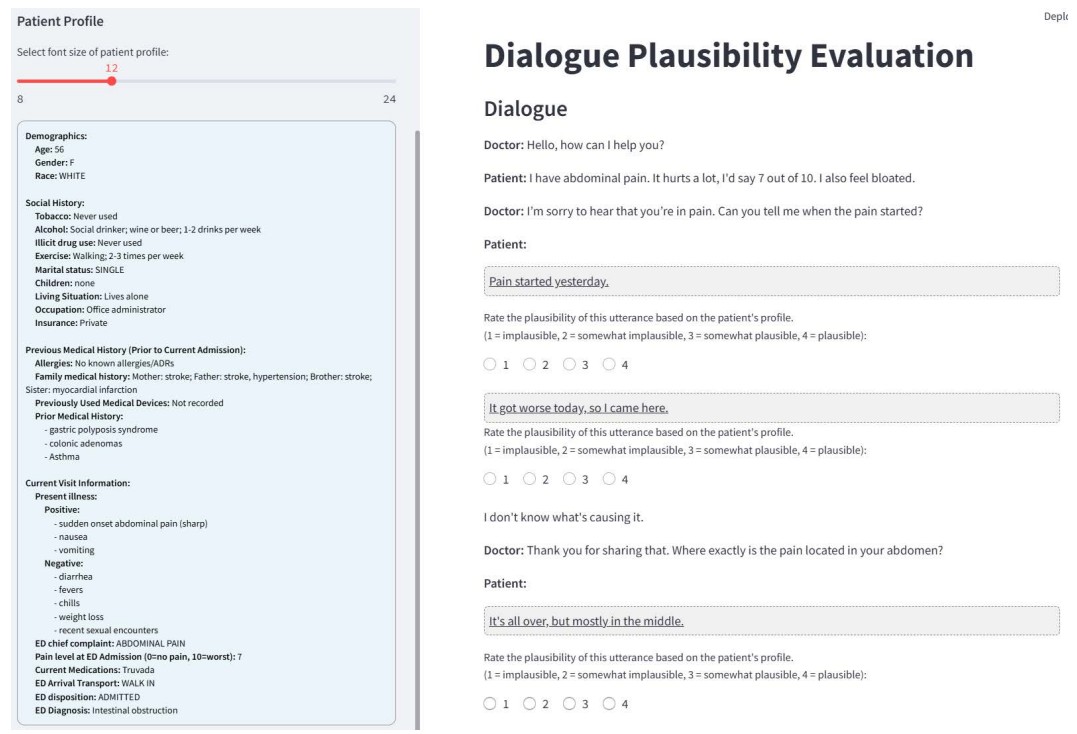

Figure E18: Screenshot of Streamlit provided to clinicians for plausibility evaluation.

## F   Experimental results

### F.1   RQ1: Do LLMs naturally reflect diverse persona traits in their responses?

#### F.1.1   Additional result of LLM evaluation

We analyze the performance of various LLMs across four persona-axis criteria, as specified in Table F8. Most models struggle to simulate negative emotions, such as distrustfulness and impatience. DeepSeek-R1-Distill-Llama-70B (*i.e.*, DeepSeek-70B) particularly underperforms on personas characterized by low recall or cognitive confusion, which often require refusing to provide a clear answer. These limitations may stem from training strategies that prioritize accurate, helpful responses or emphasize safety, avoiding potentially harmful outputs.

Table F8: Persona fidelity evaluation of various LLMs across four criteria, Personality, Language, Recall, and Confused. Each criteria evaluate the fidelity of each axis in 4 point scale. Detailed results are shown for each type.

| | Personality | | | | | | Language | | | Recall | | Confused |
|---|---|---|---|---|---|---|---|---|---|---|---|---|
| | Neutral | Distrust | Impatient | Overanxious | Overly positive | Verbose | A | B | C | High | Low | High |
| Gemini-2.5-Flash | 4.00 | 4.00 | 3.76 | 4.00 | 4.00 | 3.88 | 3.60 | 3.38 | 3.65 | 3.98 | 3.31 | 3.38 |
| GPT-4o mini | 4.00 | 2.76 | 2.76 | 4.00 | 4.00 | 4.00 | 3.43 | 3.41 | 3.84 | 3.98 | 3.59 | 3.88 |
| DeepSeek-R1-Distill-Llama-70B | 4.00 | 4.00 | 3.41 | 4.00 | 3.94 | 3.88 | 3.34 | 3.62 | 3.81 | 3.94 | 2.92 | 2.50 |
| Qwen2.5-72B-Instruct | 4.00 | 1.71 | 2.18 | 4.00 | 4.00 | 4.00 | 3.43 | 3.65 | 4.00 | 4.00 | 3.25 | 3.50 |
| Llama-3.3-70B-Instruct | 4.00 | 4.00 | 4.00 | 3.88 | 3.94 | 3.71 | 3.31 | 3.15 | 3.77 | 4.00 | 3.57 | 4.00 |
| Llama-3.1-70B-Instruct | 4.00 | 2.71 | 3.29 | 4.00 | 3.94 | 4.00 | 3.26 | 3.38 | 3.94 | 4.00 | 3.25 | 4.00 |
| Llama-3.1-8B-Instruct | 4.00 | 2.24 | 3.18 | 4.00 | 4.00 | 3.82 | 3.29 | 3.15 | 3.45 | 3.98 | 3.43 | 4.00 |
| Qwen2.5-7B-Instruct | 4.00 | 1.88 | 1.94 | 4.00 | 3.62 | 4.00 | 3.46 | 3.74 | 3.26 | 3.98 | 2.67 | 3.50 |

Table F9 presents overall consultation statistics between the doctor LLM and PATIENTSIM, including differential diagnosis (DDx) accuracy. While DDx performance does not directly measure PATIENTSIM 's capabilities, it reflects how different patient personas influence consultation complexity. After each dialogue, the doctor LLM is prompted to provide its top five differential diagnoses. This prediction is considered correct if the ground-truth diagnosis appeared in this list. To ensure consistent evaluation despite free-text outputs, we use the prompt shown in Figure F19.

On average, the doctor LLM completed the consultation and provided a final DDx within 15 turns. For personas such as verbose or advanced language proficiency, where patients provided more detailed and structured information, the model reached conclusions more quickly. Across all settings, the doctor LLM followed instructions to ask concise, focused questions, typically within three sentences. The length of PATIENTSIM 's responses differed significantly by persona. In particular, verbose and advanced personas produced substantially longer utterances, consistent with their tendency to offer elaborate or highly articulate explanations.

Table F9: Overall statistics of consultations between the doctor LLM and PATIENTSIM based on Llama 3.3 70B. # of Turns refers to the average number of dialogue turns. # Sents/Utt and # Words/Sent indicate the average number of sentences per utterance and number of words per sentence, respectively, for both the doctor LLM and PATIENTSIM. DDx represents the differential diagnosis accuracy of the doctor LLM. Results are averaged over 108 distinct consultations.

| Persona axis | Category | # of Turns | Doctor LLM | | PATIENTSIM | | DDx |
|---|---|---|---|---|---|---|---|
| | | | # Sents/Utt | # Words/Sent | # Sents/Utt | # Words/Sent | |
| Personality | neutral | 15.83 | 2.18 | 11.95 | 2.69 | 8.61 | 0.71 |
| | distrustful | 15.53 | 2.81 | 13.90 | 3.21 | 10.94 | 0.65 |
| | impatient | 13.29 | 2.27 | 12.92 | 3.02 | 8.56 | 0.59 |
| | overanxious | 13.82 | 2.39 | 14.02 | 3.07 | 13.96 | 0.94 |
| | overly positive | 13.56 | 2.18 | 12.87 | 2.74 | 12.15 | 0.38 |
| | verbose | 10.71 | 2.37 | 14.86 | 8.21 | 27.63 | 0.59 |
| Language proficiency | basic | 16.06 | 2.31 | 12.21 | 4.20 | 4.02 | 0.66 |
| | intermediate | 13.29 | 2.37 | 13.20 | 3.83 | 10.99 | 0.67 |
| | advanced | 12.39 | 2.40 | 14.78 | 3.17 | 27.01 | 0.61 |
| Medical history recall level | high | 14.40 | 2.33 | 13.41 | 3.62 | 13.75 | 0.68 |
| | low | 13.39 | 2.39 | 13.24 | 3.91 | 12.86 | 0.61 |
| Cognitive confusion level | high | 16.62 | 2.26 | 11.68 | 3.03 | 6.98 | 0.62 |
| | normal | 13.71 | 2.37 | 13.46 | 3.82 | 13.84 | 0.65 |

DDx performance varies most across the personality axis. The model shows a notable decline under the overly positive persona, likely due to PATIENTSIM downplaying symptoms, which led to less serious diagnoses. The impatient persona, marked by irritability and uncooperative behavior, and the verbose persona, with excessive or sometimes irrelevant detail, also hinder diagnostic accuracy. In contrast, the medical history recall level has a more limited impact, as present symptoms are often sufficient for DDx even without detailed historical information. Regarding cognitive confusion, direct comparisons between normal and high confusion levels should be interpreted cautiously, since other traits remain uncontrolled. Nonetheless, compared to the neutral baseline (DDx = 0.71), performance drops to 0.62 for highly confused patients, highlighting the diagnostic challenges they present.

---

**Prompt template for evaluating DDx performance**

Your task is to evaluate whether the true diagnosis is included in the predicted differential diagnoses. The predicted diagnosis can be more specific or detailed than the true diagnosis (*e.g.*, "Small Bowel Obstruction" for "Bowel Obstruction" or "Acute Pyelonephritis" for "Pyelonephritis" is acceptable), but it must not be broader than the ground truth (GT). A broader diagnosis (*e.g.*, "Pulmonary problem" for "Pneumonia') is considered incorrect. Answer with Y or N only, without further explanation.
Predicted differential diagnoses: {ddx} True diagnosis: {ans} Answer [Y/N]:

---

Figure F19: Prompt template for evaluating differential diagnosis accuracy.

### F.1.2 Additional result of human evaluation

For human evaluation, clinicians conducted consultations in total of 108 virtual patients served through PATIENTSIM, and then submitted top 3 differential diagnoses and a survey about PATIENTSIM 's overall quality. Table F10 shows overall consultation statistics between the clinician and PATIENTSIM, including DDx accuracy. Both the clinicians and the doctor LLM (from Table F9) interact with the same version of PATIENTSIM, using identical patient information and persona combinations, enabling direct comparison. Clinicians tend to ask more concise and direct questions, averaging 1.8 sentences per utterance compared to 2.4 for the LLM, and 9.1 words per sentence versus 13.3. Rather than relying on longer utterances, clinicians gather information through a greater number of shorter turns. This brevity also prompts PATIENTSIM to respond more concisely. While DDx trends align with those observed with the doctor LLM, clinicians consistently achieve more accurate diagnoses. These results highlight a notable gap in history-taking and clinical reasoning capabilities between human clinicians and the doctor LLM. In Figure F20, we provide an example of the consultation across the various persona.

Table F10: Overall statistics of consultations between clinicians and PATIENTSIM based on Llama 3.3 70B. # of Turns refers to the average number of dialogue turns. # Sents/Utt and # Words/Sent indicate the average number of sentences per utterance and number of words per sentence, respectively, for both clinicians and PATIENTSIM. DDx represents the differential diagnosis accuracy of the clinicians. Results are averaged over 108 distinct consultations.

| | | | Clinician | | PATIENTSIM | | |
|---|---|---|---|---|---|---|---|
| Persona axis | Category | # of Turns | # Sents/Utt | # Words/Sent | # Sents/Utt | # Words/Sent | DDx |
| Personality | neutral | 21.17 | 1.50 | 8.76 | 2.57 | 7.66 | 0.83 |
| | distrust | 17.82 | 1.99 | 8.72 | 3.12 | 11.66 | 0.71 |
| | impatient | 18.88 | 1.80 | 9.83 | 2.89 | 9.04 | 0.76 |
| | overanxious | 16.29 | 1.97 | 9.38 | 2.97 | 13.80 | 0.88 |
| | overly positive | 20.25 | 1.72 | 8.51 | 2.59 | 10.83 | 0.69 |
| | verbose | 13.35 | 1.93 | 9.46 | 7.85 | 13.82 | 0.71 |
| Language proficiency | basic | 21.40 | 1.62 | 8.80 | 3.81 | 4.21 | 0.74 |
| | intermediate | 17.19 | 1.80 | 8.92 | 3.64 | 10.66 | 0.83 |
| | advanced | 15.77 | 1.99 | 9.66 | 3.33 | 18.82 | 0.71 |
| Medical history recall level | high | 18.05 | 1.79 | 8.77 | 3.53 | 10.92 | 0.77 |
| | low | 18.25 | 1.81 | 9.45 | 3.69 | 10.91 | 0.76 |
| Cognitive confusion level | high | 22.12 | 1.48 | 9.00 | 3.10 | 5.88 | 0.88 |
| | normal | 17.83 | 1.82 | 9.10 | 3.65 | 11.32 | 0.76 |

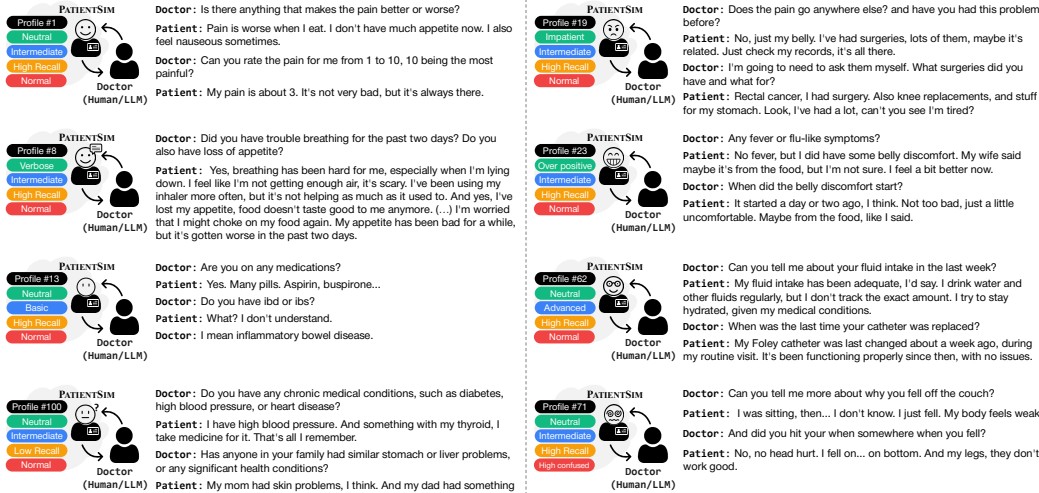

Figure F20: Example consultations for various persona options. Each simulation begins with a doctor's question, where the doctor has access only to the patient's basic information, not their symptoms. The doctor may be a real clinician or an LLM. The examples shown are mid-dialogue snippets selected to highlight the patient's persona.

Table F11: Gwet's $AC_1$ and $AC_2$ agreement between clinician and Gemini-2.5-Flash evaluation with 95% confidence intervals estimated via 1,000 bootstrap iterations.

| Metric | Gwet $AC_1$ (95% CI) | Gwet $AC_2$ (95% CI) |
|---|---|---|
| Personality | 0.897 (0.830, 0.949) | 0.957 (0.919, 0.987) |
| Language proficiency | 0.347 (0.218, 0.471) | 0.818 (0.745, 0.876) |
| Medical history recall level | 0.693 (0.585, 0.786) | 0.916 (0.865, 0.957) |
| Cognitive confusion level | 1.000 (1.000, 1.000) | 1.000 (1.000, 1.000) |
| Realism | 0.321 (0.211, 0.437) | 0.884 (0.861, 0.906) |

Table F11 presents the agreement between human evaluations and Gemini-2.5-Flash evaluations across five criteria on the same set of conversations, supporting the reliability of using Gemini for automatic evaluation. After each consultation session with PATIENTSIM, clinicians rated its overall quality for each case. To ensure a fair comparison, Gemini was provided with the same conversation logs (between the clinician and PATIENTSIM) and asked to rate the same criteria on a 4-point scale (Appendix D.2.1). When comparing the agreement between clinician ratings and Gemini ratings using Gwet's $AC_1$, we observed high agreement in personality and cognitive confusion level ($AC_1$ > 0.8), and moderate agreement in recall level ($AC_1$ = 0.693). However, agreement is relatively lower in language proficiency and realism. While $AC_1$ is designed for nominal (unordered) data and emphasizes exact matches, it can be overly strict for ordinal data. To better reflect the ordinal nature of the 4-point scale, we also computed Gwet's $AC_2$, which is more appropriate for ordered categories. Using $AC_2$, we observed high agreement ($AC_2$ > 0.8) across all criteria, indicating strong consistency between clinician and Gemini evaluations when the ordinal structure of the ratings is taken into account. These results indicate that the automatic evaluations align well with human judgment across all five criteria, with particularly strong alignment on the personality and confusion axis, even under the stricter $AC_1$ metric.

## F.2 RQ2: Do LLMs accurately derive responses based on the given profile?

Table F12 presents a detailed sentence-level factuality evaluation across eight LLMs. This table highlights that larger LLMs tend to generate more factually consistent clinical content, with higher rates of supported and entailed statements and fewer contradictions. While smaller models remain competitive, they show a slightly higher tendency to introduce contradictory information.

Table F12: Sentence-level factuality evaluation across eight LLMs, as assessed by Gemini-2.5-Flash (without normalization). # of Utter and # of Sent refer to the total number of model utterances and sentences, respectively. # of Info denotes the number of sentences categorized as informational. refer to sentences that relate to at least one item in the given profile. Unsupported statements include at least one piece of information that is not explicitly mentioned in the profile. Entail and Contradict are subsets of the supported statements.

| Model | # of Utter | # of Sent | # of Info | # of Supported | # of Unsupported | # of Entail | # of Contradict |
|-------|-----------|-----------|-----------|----------------|------------------|-------------|-----------------|
| Gemini-2.5-Flash | 889 | 2,286 | 2,220 | 1,695 | 705 | 1,659 | 36 |
| GPT-4o mini | 786 | 1937 | 1,852 | 1,331 | 795 | 1,287 | 44 |
| DeepSeek-R1-Distill-Llama-70B | 806 | 1,657 | 1,614 | 1,225 | 679 | 1,186 | 39 |
| Qwen2.5-72B-Instruct | 824 | 1,820 | 1,774 | 1,201 | 839 | 1,146 | 55 |
| Llama- 3.3-70B-Instruct | 806 | 2,180 | 2,087 | 1,654 | 817 | 1,623 | 31 |
| Llama-3.1-70B-Instruct | 699 | 1,946 | 1,842 | 1,493 | 745 | 1,438 | 55 |
| Llama-3.1-8B-Instruct | 742 | 1,774 | 1,672 | 1,284 | 826 | 1,210 | 74 |
| Qwen2.5-7B-Instruct | 877 | 1,579 | 1,558 | 1,092 | 712 | 1024 | 68 |

Llama 3.3 70B generated the fewest contradictory responses, leading us to select it as our final model. To better understand these contradiction errors, we analyzed them in detail in Table F13. The most common contradictions occurred when a patient's pain level was recorded as zero. This often happened when pain was the reason for admission but had subsided by the time of assessment or when patients presented with symptoms not typically associated with pain, such as weakness or neurological issues. In the former case, for example, patients admitted for chest or abdominal pain might later report "no pain" or "not severe," which the model may classified as contradictory. In the present illness section, contradictions frequently involved inconsistencies in symptom onset. A patient might describe a symptom as their first experience or as starting suddenly, while their medical record indicates it began days or weeks earlier. For example, a symptom that began two weeks ago could reasonably be described as "sudden" if it started abruptly at that time or as "new" if the patient had never experienced it before. Such descriptions, while potentially appearing contradictory, are often valid depending on the patient's perspective. However, due to subtle differences in wording, the model flagged these as contradictions. Some contradictions stemmed from structural issues in the patient data records. For example, a patient might be listed as widowed in the marital status section but described as caring for a spouse in the living situation section. Discrepancies also appeared in medication and medical device listings, where items mentioned in the patient's history were missing from structured fields. Overall, these contradictions were minor and not clinically significant. Given that the profiles were designed to include detailed answers to common clinical questions, a sufficiently capable model with strong contextual understanding would likely be able to generate responses with minimal contradictions. These findings suggest the potential effectiveness of our approach.

Table F13: Error analysis of sentence-level factuality evaluation. Distribution if profile categories most frequently contradicted by Llama 3.3 70B.

| Profile category | Count |
|------------------|-------|
| Pain level | 8 |
| Present illness | 6 |
| Marital status | 6 |
| Current medications | 3 |
| Medical devices | 3 |
| ED chief complaint | 2 |
| ED arrival transport | 2 |
| Alcohol | 1 |
| Family medical history | 1 |

## F.3 RQ3: Can LLMs reasonably fill in the blanks?

Table 4 in the main paper presents each labeler's plausibility ratings for answers about unspecified information, along with inter-rater agreement metrics. Here, we evaluate the agreement between Gemini-2.5-Flash and the human labelers by having Gemini-2.5-Flash rate the same set of PATIENTSIM's responses. For each labeler, agreement with the LLM was computed over an average of 615 responses. Across all labelers, we observe Gwet's $AC_1$ agreement scores above 0.8, demonstrating the reliability of Gemini's automatic plausibility assessments.

Table F14: Plausibility scores for unsupported sentences in patient responses, labeled by four clinicians. Each clinician annotated 39 consultation, around 615 sentences per each. We automatically annotate the same sentences using Gemini-2.5-Flash, and measure the clinician-LLM agreement measured by Gwet's $AC_1$ with 95% confidence intervals estimated via 1,000 bootstrap iterations.

|  | Clinician A | Clinician B | Clinician C | Clinician D |
|---|---|---|---|---|
| Gemini-2.5-Flash | 0.944 (0.923, 0.960) | 0.945 (0.926, 0.961) | 0.964 (0.947, 0.977) | 0.883 (0.857, 0.907) |

## F.4 Ablation study

### F.4.1 Validation of sentence-level classification

We validated the performance of an LLM sentence classifier in detecting supported and unsupported statements using 10 distinct profiles. From 10 consultations, we extracted 411 sentences, which were manually annotated by the author on a sentence-by-sentence basis. Of these, 93% (382 sentences) were classified as informational. These informational sentences were further annotated to determine: 1) related profile items (*e.g.*, age, gender), 2) whether each sentence was entailed or contradicted by the profile, and 3) whether the sentence contained information not explicitly mentioned in the profile. Manual annotations are compared against predictions from Gemini-2.5-Flash (Gemini) and GPT-4o-1120 (GPT-4o) to evaluate classification performance across four categories:

- Sentence category classification: Identifies whether a sentence is informational or non-informational.
  - Acc (%): Proportion of correct classifications.
  - Recall (%): Proportion of true informational sentences correctly identified.
- Detection of related profile items: Assesses the model's ability to identify correct profile items related to each sentence, measured by:

$$P_{\text{item}} = \frac{|\text{Pred}_{\text{item}} \cap \text{GT}_{\text{item}}|}{|\text{Pred}_{\text{item}}|}, \quad R_{\text{item}} = \frac{|\text{Pred}_{\text{item}} \cap \text{GT}_{\text{item}}|}{|\text{GT}_{\text{item}}|}, \quad F1_{\text{item}} = \frac{2 \cdot P_{\text{item}} \cdot R_{\text{item}}}{P_{\text{item}} + R_{\text{item}}} \quad (5)$$

  where $\text{Pred}_{\text{item}}$ is the set of profile items predicted by the model, and $\text{GT}_{\text{item}}$ is the set of ground truth items annotated by the human.
- Entailment evaluation for detected items: Measures accuracy in classifying entailment or contradiction for correctly detected profile items.
  - $Acc_{\text{val}}$ (%): Proportion of correct entailment/contradiction labels among overlapping keys.
- Unsupported Information Detection: Evaluates the model's ability to identify sentences with information not explicitly in the profile, measured by:

$$P_{\text{unsupp}} = \frac{\text{TP}_{\text{unsupp}}}{|\text{Pred}_{\text{unsupp}}|}, \quad R_{\text{unsupp}} = \frac{\text{TP}_{\text{unsupp}}}{|\text{GT}_{\text{unsupp}}|}, \quad F1_{\text{unsupp}} = \frac{2 \cdot P_{\text{unsupp}} \cdot R_{\text{unsupp}}}{P_{\text{unsupp}} + R_{\text{unsupp}}} \quad (6)$$

  where $\text{TP}_{\text{unsupp}}$ is the number of unsupported sentences correctly identified by the model, $\text{Pred}_{\text{unsupp}}$ is the set of sentences predicted as unsupported by the model, and $\text{GT}_{\text{unsupp}}$ is the set of ground truth unsupported sentences.

Table F15 summarizes the performance of Gemini and GPT-4o. Gemini outperforms GPT-4o overall, particularly in recall, despite GPT-4o showing slightly higher precision in detecting related profile items and unsupported information. In medical applications, recall is prioritized over precision to minimize missed detections, which could have critical consequences. As both models achieve precision above 0.8, indicating robust performance, Gemini's superior recall makes it the preferred evaluator.

Table F15: Comparison of sentence-level evaluation metrics between Gemini-2.5-Flash and GPT-4o.

| Metric | Gemini-2.5-Flash | Gpt-4o |
|---|---|---|
| *Sentence category classification* | | |
| Acc (%) | 0.96 | 0.94 |
| Recall (%) | 0.99 | 0.98 |
| *Detect related profile items* | | |
| $P_{key}$ | 0.90 | 0.92 |
| $R_{key}$ | 0.96 | 0.94 |
| $F1_{key}$ | 0.92 | 0.92 |
| *Entailment evaluation* | | |
| $Acc_{val}$ | 0.98 | 0.97 |
| *Detect unsupported information* | | |
| $P_{unsupp}$ | 0.84 | 0.89 |
| $R_{unsupp}$ | 0.86 | 0.64 |
| $F1_{unsupp}$ | 0.84 | 0.74 |

## F.4.2  Ablation study on doctor LLM

To evaluate the ability of LLMs as the doctor, to elicit and assess patient responses, we conducted an ablation study focusing on the doctor LLM's capacity to extract information from diverse patient personas. While the main study assessed patients' ability to provide accurate information, this study examines how effectively the doctor LLM gathers information across varied patient profiles. We measured three metrics: *ICov*, *ICon*, and their product (*Weighted ICon*), to assess the amount and consistency of information extracted. These metrics were calculated for 108 patients, each assigned one of 37 distinct personas randomly. In this study, we varied only the doctor model, testing Gemini-2.5-Flash, GPT-4o mini, and Llama 3.3 70B, while fixing the PATIENTSIM LLM backbone to Llama 3.3 70B. In Table F16, GPT-4o mini achieved the highest *ICov* and *Weighted ICon* scores, demonstrating superior performance in extracting information. Consequently, we selected GPT-4o mini as the doctor LLM for the automatic evaluation of PATIENTSIM.

Table F16: Dialogue-level factuality evaluation across Social History (Social), Previous Medical History (PMH), and Current Visit Information (Current Visit). *Information Consistency (ICon)* is rated on a 4-point scale by Gemini-2.5-Flash, and *Weighted ICon* represents *Information Coverage-Weighted Consistency*, reflecting both coverage and consistency.

| | Information Coverage (ICov) (%) | | | Infomation Consistency (ICon) | | | Weighted ICon | | | |
|---|---|---|---|---|---|---|---|---|---|---|
| | Social | PMH | Current Visit | Social | PMH | Current Visit | Social | PMH | Current Visit | Avg. |
| Gemini-2.5-Flash | 0.34 | 0.72 | 0.82 | 3.74 | 3.18 | 2.98 | 1.27 | 2.29 | 2.44 | 2.00 |
| GPT-4o mini | 0.44 | 0.74 | 0.86 | 3.78 | 3.03 | 2.92 | 1.66 | 2.24 | 2.51 | 2.14 |
| Llama-3.3-70B-Instruct | 0.31 | 0.54 | 0.79 | 3.71 | 3.14 | 2.95 | 1.15 | 1.70 | 2.33 | 1.73 |

## F.4.3  Analysis of the impact of high confusion on other persona traits

We imposed certain limitations on the highly confused patient category because the typical behaviors observed in such patients often overlap or conflict with other persona dimensions. Specifically, highly confused patients tend to: 1) misunderstand doctors' questions and repeat themselves, 2) fail to recall past events accurately, or 3) appear overly anxious or even aggressive due to being emotionally overwhelmed. These behaviors coincide, respectively, with low Language proficiency, poor Recall ability, and extreme Personality traits. As a result, it becomes difficult to distinguish whether a given behavior stems from the patient's confusion or from their underlying persona attributes. For example, a confused patient with high language proficiency may appear to have poor language skills simply due to confusion, and a patient with naturally low recall may be indistinguishable from the one whose memory is temporarily impaired by confusion.

To investigate this further, we consulted two medical experts, each with over 10 years of clinical experience, including an ER doctor with 13 years of experience. They interacted with highly confused patients who were assigned all possible combinations of persona attributes across other axes (*i.e.*, personality, language, recall). Both experts independently concluded that the confused state often overshadows traits from other persona axes, making it extremely difficult to evaluate each dimension.

To support these observations, we conducted an additional experiment. We selected a subset of test cases consisting of "normal" patients (Normal row in Table F17) and modified only their confusion level to "highly confused," (Highly Confused row in Table F17) while keeping all other attributes unchanged. We then re-simulated these conversations using PATIENTSIM with Llama 3.3 70B, and evaluated them using Gemini-2.5-Flash. As shown in the Table F17, while the Confused and Realism dimension retained reasonably high scores, indicating effective confusion simulation, Personality, Language, and Recall scores dropped significantly despite those remaining fixed. This suggests that confusion distorts the expression of other traits, making it difficult to interpret model behavior or ensure reliable evaluations across all dimensions.

Table F17: Persona fidelity scores when increasing the confusion level of normal patients to highly confused while keeping other persona attributes fixed.

| Metric | Personality | Language | Recall | Confused | Realism | Avg |
|---|---|---|---|---|---|---|
| Normal | 3.90 | 3.42 | 3.79 | 4.00 | 3.97 | 3.70 |
| Highly Confused | 3.62 | 3.17 | 3.44 | 3.64 | 3.82 | 3.54 |

## G   Responsible use and limitations

Our open-source patient simulator framework provides a safe, privacy-preserving environment to evaluate clinical LLMs through realistic interactions without real patient data. While its 37 predefined personas offer diverse scenarios, they may not fully cover the variability of real-world clinical settings. Additionally, simulated conversations cannot capture non-verbal cues, and over-reliance on the simulator may limit the assessment of practical clinical skills. The simulator is not intended for developing clinical decision-making tools for real patient care without rigorous clinical oversight, as it is designed solely for educational and research purposes. Acknowledging these limitations and incorporating expert feedback are essential for its effective use.

