# OpenReview forum: "PatientSim: A Persona-Driven Simulator for Realistic Doctor-Patient Interactions"
_NeurIPS.cc/2025/Datasets_and_Benchmarks_Track — NeurIPS 2025 Datasets and Benchmarks Track spotlight_

### Official Review · Reviewer_RK5W · 2025-06-07

**Rating:** 5
**Confidence:** 4

**Summary:**

The paper presents PATIENTSIM, a patient simulator engineered to simulate realistic and diverse doctor-patient interactions. It constructs patient personas utilizing data from the MIMIC-IV and MIMIC-ED datasets, encompassing four dimensions: personality, language proficiency, medical history recall, and cognitive disorientation. The primary objective of PATIENTSIM is to augment the realism of clinical training and the evaluation of LLM-driven doctor agents. Eight LLMs were assessed for persona fidelity and factual accuracy, with Llama 3.3 emerging as the most proficient model. PATIENTSIM has been validated by clinicians, demonstrating its realism, accuracy, and educational value.

**Dataset Code Accessibility:**

Yes

**Dataset Code Comments:**

The dataset and source code are available on Kaggle and the anonymous GitHub platform.

**Ethical Considerations:**

No, there are no or only very minor ethics concerns

**Final Justification:**

The response from authors effectively addressed my concerns and highlighted an oversight on my previous review. I increase my score to 5 based on the discussion.

**Limitations Weaknesses:**

- The emphasis on single-session emergency department visits solely for history-taking significantly restricts clinical realism. Real medical practice encompasses physical examinations, diagnostic tests, and longitudinal care—all explicitly excluded from PATIENTSIM. While the authors justify this approach by highlighting the 80% of diagnoses made through history-taking alone, it presents an idealized scenario rarely encountered in clinical settings.
- The system’s scope is limited to only five diseases, and only textual information is included. This narrow focus undermines its utility as a comprehensive training tool. Furthermore, the text-only interaction format disregards crucial non-verbal communication elements.
- The clinical validation process involved only four general practitioners from a single recruitment source. For a tool intended for widespread educational use, this sample size is insufficient to establish generalizability. Additionally, all patient data originates from a single US hospital, raising concerns about the representativeness of demographic and practice patterns.
- There are certain artificial limitations, such as the restriction of highly confused patients to neutral personality, intermediate language proficiency, and high recall. The reliability and expertise of these limitations are uncertain.

**Strengths Contributions:**

- PATIENTSIM introduces a sophisticated four-axis persona system. This framework captures clinically relevant patient characteristics that directly impact consultation quality, as validated by medical experts. The systematic combination of personality types, language proficiency levels, recall abilities, and confusion states creates realistic patient diversity.
- The integration of MIMIC-IV and MIMIC-ED datasets provides authentic clinical foundations. The authors’ hybrid approach combining structured data with clinical notes ensures both accuracy and narrative depth. The preprocessing pipeline, including alignment verification and context-aware imputation of missing values, demonstrates meticulous attention to data quality.
- The three-tiered evaluation approach is noteworthy. Sentence-level factual accuracy assessment with detailed natural language inference evaluation. Dialogue-level information coverage and consistency metrics. Clinical plausibility ratings for information not explicitly present in patient profiles.

---

> ### Author Rebuttal · Authors · 2025-07-30
>
> We sincerely appreciate your thorough review of our paper and your constructive suggestions. Our responses to your comments are provided in the following Q&A pairs. We will revise the manuscript to improve clarity in response to the provided feedback.
>
> **C1: The emphasis on single-session emergency department visits solely for history-taking significantly restricts clinical realism.**
>
> **A1:**  We acknowledge that integrating additional tests or longitudinal care would enhance clinical applicability. However, we want to emphasize that doing so is practically infeasible. Our system is designed to allow the doctor LLM to ask open-ended questions freely. In this setup, incorporating diagnostic test data becomes technically infeasible, as the LLM may request tests that are unavailable for a given case. Currently, we lack reliable methods to generate realistic and case-consistent synthetic test results that can cover the full range of possible queries. Moreover, only a small portion of patients (e.g. ED visits) include lab or imaging data prior to any treatments or prescriptions. Since our goal is to model patients’ initial state at admission, we selected test data prior to any clinical interventions that could alter their presentation. In our pre-processed subset (Sec. A.2), only ~2% of patients have lab/chart events and ~3.3% have radiology reports. Thus, most records lack the early diagnostic information needed for robust test-based simulations.
>
> Extending the simulation to include longitudinal care introduces additional challenges. Incorporating follow-up strategies risks the doctor LLM recommending treatments not in the database. Current simulation technologies are unable to reliably model these counterfactual, longitudinal clinical scenarios. Patient outcomes can vary substantially depending on numerous factors such as comorbidities, lifestyle, and medication adherence. Predicting these trajectories is highly complex, even for experienced physicians. Attempting to model longitudinal states with existing model risks generating misleading or clinically unsafe conclusions.
>
> &nbsp;
>
> **C2: The system’s scope is limited to only five diseases, and only textual information is included.**
>
> **A2:** As discussed above (C1-A1), our study focuses on the history-taking phase of ED encounters, relying on patients’ verbal information to establish an initial DDx. Within this scope, we selected five diseases based on the following criteria: 1) high clinical prevalence and significance as common ED presentations; 2) distinct and differentiable symptom profiles suitable for diagnosis through history-taking alone; and 3) sufficient representation in the MIMIC-ED to support robust model development and evaluation. Since the target diseases must be reliably inferred from conversational information alone without any test results, this requirement narrows the range and granularity of disease categories. For example, physicians cannot reliably distinguish between myocardial infarction and angina just by talking to the patient. This selection was guided by two medical experts, including an ER doctor with 13 years of clinical experience, ensuring that the disease coverage is clinically meaningful and consistent with ED workflows.
>
> While the current disease scope is constrained by these practical considerations, PatientSim has demonstrated its potential for expansion by delivering clinically accurate and coherent patient information (Table 2). As larger and more diverse datasets become available, the simulator can be readily extended to include additional conditions, thereby broadening its applicability to a wider range of clinical scenarios.
>
> Regarding modality, we agree that incorporating multimodal data to capture non-verbal cues is a promising direction for future work. However, this area remains relatively underexplored. As Steenstra et al. [1] note, current VR patient simulations often produce machine-like responses, lack natural intonation, and fail to represent personality traits in a meaningful way. Given these limitations, the use of VR in patient simulation may still be premature. Due to these constraints, most existing research on simulating patient-doctor interactions continues to rely on text-based simulations. Within this constraint, our patient simulator uses tone indicators and punctuation, such as ellipses or emphatic phrasing, to express subtle emotional cues and enhance the realism of the simulation. For example, a line like “Look, I’ve had a lot, can’t you see I’m tired?” reflects an impatient tone, while “I was sitting, then… I don’t know. I just fell.” expresses confusion and hesitation (Figure F20).
>
> &nbsp;
>
> **C3: The clinical validation process involved only four general practitioners from a single recruitment source. Additionally, all patient data originates from a single US hospital.**
>
> **A3:** Regarding the human evaluation, we were limited to four clinicians due to budget constraints, with a total of €2,500 allocated for 45 hours of evaluation work. Despite the small sample size, we consider the evaluators’ insights to be highly valuable given their substantial clinical expertise. Two of the evaluators are licensed physicians with four years of clinical experience. The other two are also licensed physicians with six years of clinical experience and additionally hold nursing licenses, with 13 and 17 years of prior nursing practice, respectively. The clinicians were recruited via Ingedata, an AI data annotation company that serves as an intermediary between clinicians and research teams. Importantly, all four evaluators are affiliated with different hospitals or medical institutions, contributing to a wider range of clinical perspectives and helping to mitigate potential institutional bias.
>
> Notably, many prior studies in this domain have relied solely on LLM-based evaluations or involved a smaller number of human evaluators, often medical students [1,2]. This is partly due to the difficulty and cost of recruiting experienced medical professionals. In this context, and within the given constraints, our approach provides a robust and realistic human evaluation framework, grounded in clinical expertise and institutional diversity.
>
> Limited access to clinical data restricts available data sources. Simulating patients requires detailed clinical information, such as presenting symptoms, pain levels, and family medical history. These elements are critical for physicians during the initial diagnosis process, as they form the foundation of clinical reasoning and decision-making [3]. Currently, the MIMIC database is the only publicly available dataset that offers this level of clinical detail. While this limits the generalizability of our findings, it reflects an unavoidable constraint given the need for clinically rich data to ensure simulation fidelity.
>
> [1] H. Wei et al., Medco: Medical education copilots based on a multi-agent framework, ECCV 2024 Workshops, 2024.
> [2] Z. Fan et al., Ai hospital: Benchmarking large language models in a multi-agent medical interaction simulator. In COLING, 2025.
> [3] E. C. Toy et al., Case Files Emergency Medicine, 4th edition, 2017.
>
> &nbsp;
>
> **C4: There are certain artificial limitations, such as the restriction of highly confused patients.**
>
> **A4:** We imposed certain limitations on the highly confused patient category because the typical behaviors observed in such patients often overlap or conflict with other persona dimensions.
> Specifically, highly confused patients tend to: 1) Misunderstand doctors' questions and repeat themselves, 2) Fail to recall past events accurately, or 3) Appear overly anxious or even aggressive due to being emotionally overwhelmed.
>
> These behaviors coincide, respectively, with low Language proficiency, poor Recall ability, and extreme Personality traits. As a result, it becomes difficult to distinguish whether a given behavior stems from the patient's confusion or from their underlying persona attributes. For example, a confused patient with high language proficiency may appear to have poor language skills simply due to confusion, and a patient with naturally low recall may be indistinguishable from the one whose memory is temporarily impaired by confusion.
>
> To investigate this further, we consulted two medical experts, each with over 10 years of clinical experience. They interacted with highly confused patients who were assigned all possible combinations of persona attributes across other axes (i.e., language, personality, recall). Both experts independently concluded that the confused state often overshadows traits from other persona axes, making it extremely difficult to evaluate each dimension.
>
> To support these observations, we conducted an additional experiment. We selected a subset of test cases consisting of “normal” patients and modified only their confusion level to “highly confused,” keeping all other attributes unchanged. We then re-simulated these conversations using PatientSim with Llama 3.3 70B, and evaluated them using Gemini 2.5 Flash.
>
> As shown in the table, while the Confused/Realism dimension retained reasonably high scores, indicating effective confusion simulation, Personality, Language, and Recall scores dropped significantly despite those remaining fixed. This suggests that confusion distorts the expression of other traits, making it difficult to interpret model behavior or ensure reliable evaluations across all dimensions.
>
> |Metric|Personality|Language|Recall|Confused|Realism|Avg|
> |---|---|---|---|---|---|---|
> |**Original**|3.90|3.42|3.79|4.00|3.97|3.70|
> |**Reverse**|3.62|3.17|3.44|3.64|3.82|3.54|
>
> &nbsp;
>
> As discussed in **C1-C4**, while some limitations are inevitable due to practical, technical, and financial constraints, we believe our study makes a meaningful contribution. With continued advances in technology and data accessibility, this work lays the foundation for more advanced simulation research in the future.

---

> > ### Comment · Reviewer_RK5W · 2025-08-05
> > **Thanks for Response**
> >
> > Thanks for the authors' comprehensive and well-considered response, which effectively addressed my concerns and highlighted an oversight on my previous review.
> >
> >  I will increase my score to 5 and extend my best wishes for this pap

---

### Official Review · Reviewer_Hr6d · 2025-07-02

**Rating:** 4
**Confidence:** 4

**Summary:**

This paper introduces PATIENTSIM, a persona-driven patient simulator designed to generate realistic doctor-patient interactions for training and evaluating medical dialogue systems. The system combines clinical profiles derived from MIMIC-IV and MIMIC-ED datasets with 37 distinct patient personas characterized along four axes: personality, language proficiency, medical history recall level, and cognitive confusion level. The authors evaluate eight LLMs and select Llama 3.3 as the final model, which is further validated by four clinicians. The simulator achieves an average quality score of 3.89/4 across six evaluation criteria and demonstrates strong performance in both factual accuracy and persona consistency.

**Dataset Code Accessibility:**

Partly

**Ethical Considerations:**

No, there are no or only very minor ethics concerns

**Limitations Weaknesses:**

1 The system only covers five diseases (myocardial infarction, pneumonia, UTI, intestinal obstruction, stroke). While these are common ED presentations, this limitation significantly restricts the simulator's applicability to broader clinical scenarios.
2 The inability to capture non-verbal cues (facial expressions, body language, tone of voice) limits the realism of the simulation. As acknowledged in Section 8, these elements are crucial in real clinical encounters.
3 The focus on single ED visits without modeling longitudinal patient states or treatment effects reduces clinical applicability for many real-world scenarios requiring follow-up care.
4 While the four clinicians provide valuable validation, the evaluation scale is relatively small. Given the importance of clinical validity, a larger and more diverse group of evaluators would strengthen the findings.
5 The paper mentions avoiding overlap between confusion and other axes (page 4, line 165-167), but the rationale for limiting highly confused patients to specific persona combinations is not fully justified. This design choice may miss important real-world patient presentations.

**Strengths Contributions:**

1 The paper presents a well-motivated approach to patient simulation that addresses a critical gap in medical AI evaluation. Unlike previous work that focuses solely on factual accuracy, PATIENTSIM incorporates diverse patient personas that reflect real-world consultation complexities.
2 The four-axis persona design (personality, language proficiency, recall level, confusion level) is grounded in medical literature and expert guidance. This multi-dimensional approach captures important factors that influence doctor-patient communication quality, going beyond simplistic patient representations in prior work.
3 The paper employs both automated and human evaluation across three research questions, assessing persona fidelity, factual accuracy, and clinical plausibility. The sentence-level factuality evaluation framework (Figure 2) is particularly thorough and well-designed.
4 The system demonstrates high performance across all evaluation metrics, with Llama 3.3 achieving 98.1% entailment accuracy for supported statements and 3.96/4 plausibility for unsupported statements. The human evaluation by four clinicians provides additional validation with high inter-annotator agreement (Gwet's AC1 > 0.85).

---

> ### Author Rebuttal · Authors · 2025-07-30
>
> We are truly grateful for the time and effort you have dedicated to carefully reviewing our manuscript and for providing constructive feedback. Our replies to each of your comments are provided in the following Q&A pairs. We will revise the manuscript to improve clarity in response to the provided feedback.
>
> **C1: The system only covers five diseases.**
>
> **A1:** Our study focuses on the history-taking phase of ED encounters, during which physicians typically rely on patients’ verbal information to establish an initial DDx. The reason for focusing solely on doctor-patient conversation, and not incorporating lab tests or images in this work is twofold: 1) In the MIMIC dataset, only a small fraction of ED visits include lab or radiology data before any treatment or prescriptions. Since our goal is to model patients’ initial state at admission, we selected test data prior to any clinical interventions that might alter patient presentation. In our pre-processed subset (Section A.2), only ~2% of patients have lab/chart events and ~3.3% have radiology reports. 2) If we incorporate lab tests in our simulation, the doctor LLM might seek test results that do not exist in the MIMIC. Bridging this gap would require a reliable test result prediction model, but such technology is not yet sufficiently mature. Therefore, our simulator operates exclusively in a dialogue-based setting.
>
> Within this scope, we selected five diseases based on the following criteria: 1) high clinical prevalence and significance as common ED presentations; 2) distinct and differentiable symptom profiles suitable for diagnosis through history-taking alone; and 3) sufficient representation in the MIMIC-ED to support robust model development and evaluation. Since the target diseases must be reliably inferred from conversational information alone without any test results, this requirement narrows the range and granularity of disease categories. For example, physicians cannot reliably distinguish between myocardial infarction and angina just by talking to the patient. This selection was guided by two medical experts, including an ER doctor with 13 years of clinical experience, ensuring that the disease coverage is clinically meaningful and consistent with ED workflows.
>
> While the current disease scope is constrained by these practical considerations, PatientSim has demonstrated its potential for expansion by delivering clinically accurate and coherent patient information (Table 2). As larger and more diverse datasets become available, the simulator can be readily extended to include additional conditions, thereby broadening its applicability to a wider range of clinical scenarios.
>
> &nbsp;
>
> **C2:The inability to capture non-verbal cues limits the realism of the simulation.**
>
> **A2:** Incorporating multimodal data to capture non-verbal cues, potentially through VR or other simulation technologies, is a promising direction for future work. However, this area remains relatively underexplored. As Steenstra et al. [1] note, current VR patient simulations often produce machine-like responses, lack natural intonation, and fail to represent personality traits in a meaningful way. Given these limitations, the use of VR in patient simulation may still be premature. Due to these constraints, most existing research on simulating patient-doctor interactions continues to rely on text-based simulations. Within this constraint, our patient simulator uses tone indicators and punctuation, such as ellipses or emphatic phrasing, to express subtle emotional cues and enhance the realism of the simulation. For example, a line like “Look, I’ve had a lot, can’t you see I’m tired?” reflects an impatient tone, while “I was sitting, then… I don’t know. I just fell.” expresses confusion and hesitation (Figure F20).
>
> [1] Steenstra et al., Scaffolding Empathy: Training Counselors with Simulated Patients and Utterance-level Performance Visualizations, In CHI '25, 2025.
>
> &nbsp;
>
> **C3:The focus on single ED visits without modeling longitudinal patient states or treatment effects reduces clinical applicability for many real-world scenarios requiring follow-up care.**
>
> **A3:** We acknowledge that modeling longitudinal patient states and treatment effects would enhance clinical applicability. However, this remains a significant challenge due to the nature of clinical simulation setup as noted in Section 3 (line 114). Due to space constraints, we were not able to provide an extended elaboration regarding this matter in the main text, so we provide a detailed explanation here.
>
> Our system is designed to allow the doctor LLM to ask open-ended questions freely. In this setup, incorporating interventions, medications, or follow-up strategies presents a risk: the doctor LLM may suggest treatments different from the ones in the database. Current simulation technologies are unable to reliably model these counterfactual, longitudinal clinical scenarios. Patient outcomes can vary substantially depending on numerous factors such as comorbidities, lifestyle, and medication adherence. Predicting these trajectories is highly complex, even for experienced physicians. Attempting to model longitudinal clinical states with existing model risks generating misleading or clinically unsafe conclusions.
>
> While these limitations are inevitable due to practical and technical constraints, we believe our study still makes a meaningful contribution, as accurate information gathering during this phase is crucial for diagnosis and subsequent clinical decision-making.
>
> &nbsp;
>
> **C4: While the four clinicians provide valuable validation, the evaluation scale is relatively small.**
>
> **A4:** We recognize the concern about the limited scale of the human evaluation. Due to budgetary constraints, we were limited to engaging four clinicians, with a total allocation of €2,500 covering 45 hours of evaluation work. Despite the small sample size, we consider the evaluators’ insights to be highly valuable given their substantial clinical expertise. Two of the evaluators are licensed physicians with four years of clinical experience. The other two are also licensed physicians with six years of clinical experience and additionally hold nursing licenses, with 13 and 17 years of prior nursing practice, respectively. The clinicians were recruited via Ingedata, an AI data annotation company that serves as an intermediary between clinicians and research teams. Importantly, all four evaluators are affiliated with different hospitals or medical institutions, contributing to a wider range of clinical perspectives and helping to mitigate potential institutional bias.
>
> Notably, many prior studies in this domain have relied solely on LLM-based evaluations or involved a smaller number of human evaluators, often medical students [1,2]. This is partly due to the difficulty and cost of recruiting experienced medical professionals. In this context, and within the given constraints, our approach provides a robust and realistic human evaluation framework, grounded in clinical expertise and institutional diversity.
>
> [1] H. Wei et al., Medco: Medical education copilots based on a multi-agent framework, ECCV 2024 Workshops, 2024.
> [2] Z. Fan et al., Ai hospital: Benchmarking large language models in a multi-agent medical interaction simulator. In COLING, 2025.
>
> &nbsp;
>
> **C5: The paper mentions avoiding overlap between confusion and other axes (page 4, line 165-167), but the rationale for limiting highly confused patients to specific persona combinations is not fully justified.**
>
> **A5:** We imposed certain limitations on the highly confused patient category because the typical behaviors observed in such patients often overlap or conflict with other persona dimensions. Specifically, highly confused patients tend to: 1) Misunderstand doctors' questions and repeat themselves, 2) Fail to recall past events accurately, or 3) Appear overly anxious or even aggressive due to being emotionally overwhelmed.
>
> These behaviors coincide, respectively, with low Language proficiency, poor Recall ability, and extreme Personality traits. As a result, it becomes difficult to distinguish whether a given behavior stems from the patient's confusion or from their underlying persona attributes. For example, a confused patient with high language proficiency may appear to have poor language skills simply due to confusion, and a patient with naturally low recall may be indistinguishable from the one whose memory is temporarily impaired by confusion.
>
> To investigate this further, we consulted two medical experts, each with over 10 years of clinical experience, including an ER doctor with 13 years of experience. They interacted with highly confused patients who were assigned all possible combinations of persona attributes across other axes (i.e., language, personality, recall). Both experts independently concluded that the confused state often overshadows traits from other persona axes, making it extremely difficult to evaluate each dimension.
>
> To support these observations, we conducted an additional experiment. We selected a subset of test cases consisting of “normal” patients and modified only their confusion level to “highly confused,” while keeping all other attributes unchanged. We then re-simulated these conversations using PatientSim with Llama 3.3 70B, and evaluated them using Gemini 2.5 Flash.
>
> As shown in the table, while the Confused/Realism dimension retained reasonably high scores, indicating effective confusion simulation, Personality, Language, and Recall scores dropped significantly despite those remaining fixed. This suggests that confusion distorts the expression of other traits, making it difficult to interpret model behavior or ensure reliable evaluations across all dimensions.
>
> |Metric|Personality|Language|Recall|Confused|Realism|Avg|
> |---|---|---|---|---|---|---|
> |**Original**|3.90|3.42|3.79|4.00|3.97|3.70|
> |**Reverse**|3.62|3.17|3.44|3.64|3.82|3.54|

---

> > ### Comment · Reviewer_Hr6d · 2025-08-06
> >
> > Thank you for the authors' thoughtful engagement and the detailed clarifications provided in response to my questions. I appreciate the effort the authors put into addressing the concerns. However, after considering the overall contribution in terms of topic relevance, novelty, and quality, I will maintain my original score.

---

### Official Review · Reviewer_Fojh · 2025-07-02

**Rating:** 5
**Confidence:** 4

**Summary:**

The authors present the patient simulator using MIMIC dataset by adding several important perspectives including personality, cognitive level, medical history recall level and language proficiency. It covers rich perspectives leading to over 100 personas for patient going to ED.

**Dataset Code Accessibility:**

No

**Dataset Code Comments:**

There is no link available for data and code access.

**Ethical Considerations:**

No, there are no or only very minor ethics concerns

**Final Justification:**

The authors have provided great explanation on the real-world patient simulation that some data are not easily available. They have provided satisfied answers. I have updated my rating.

**Limitations Weaknesses:**

- The authors mentioned in the paper that the patient simulator will not ask lab test and imaging results, but there are lab test and radiology report in MIMIC dataset. Of course, many could be missing, but incorporating lab tests and imaging results are critical for any real-world clinical diagnosis.
- The evaluation criteria for RQ1 are too general, more details will be better to understand the changes or divergence in their personality, recall, and cognitive level. Some open-ended questions to provide details can be added to the evaluation.
- in Figure D11, how accurate for this binary classification for over 15+ different categories. As LLMs hallucinate, it will be good to test the robustness of such classifications for different LLMs and multiple times. Same for figure D12.
- In Figure 20, the doctor and patient conversations all start with doctors who seems know the patient condition already. But in a realistic scenario, the first question from doctor could be why you are coming to ED?, then the conversation follows. Can you show a full conversation with several turns?

**Strengths Contributions:**

+ it is interesting to see authors add language proficiency, medical history recall level, and level of cognitive confusion to the patient simulator. The combination of them leads to 37 distinct personas which are rich and realistic.
+ Great variety of patient profiles of a total 179 patient profiles rich in age group, race, martial status, insurance, arrival transport, pain score, and diagnosis.
+ great research questions and through evaluations based on LLM as a judge and four clinicians.

---

> ### Author Rebuttal · Authors · 2025-07-30
>
> We deeply appreciate the time and effort you dedicated to reviewing our manuscript and providing thoughtful feedback. We address your comments in the following Q&A pairs.
>
> **C1: The authors mentioned in the paper that the patient simulator will not ask lab test and imaging results, but there are lab test and radiology report in MIMIC dataset. Of course, many could be missing, but incorporating lab tests and imaging results are critical for any real-world clinical diagnosis.**
>
> **A1:** We agree that incorporating lab tests and imaging results would enhance the clinical realism of our simulations. However, we want to emphasize that doing so is practically infeasible.
> While the MIMIC database does contain laboratory and radiology data, only a small portion of patients (e.g. ED visits) includes such information prior to any treatments or prescriptions, a critical constraint for our use case. Since our objective is to model patients’ initial state at admission, before any clinical interventions alter their presentation, we must isolate data that reflects this early phase. In our pre-processed subset, which links MIMIC-ED encounters with MIMIC-IV discharge notes, only about 2% of patients (i.e. ED visits) include lab or chart events, and just 3.3% contain at least one associated radiology report prior to any clinical interventions. Note that the radiology data in MIMIC-IV is limited to chest X-rays, further limiting the diagnostic breadth. As a result, the majority of patient records lack the comprehensive early diagnostic information needed to support robust test-based simulations.
>
> Moreover, a key challenge lies in the simulation setting: our system is designed to allow the doctor LLM to ask open-ended questions freely. In this setup, incorporating diagnostic test data becomes technically infeasible, as the LLM may request tests that are unavailable for a given case. At present, we lack reliable methods to generate realistic and case-consistent synthetic test results that can cover the full range of possible queries.
>
> Despite this limitation, PATIENTSIM still provides substantial value by supporting realistic, interactive history-taking based on real ED encounters, an essential component of the diagnostic process.
>
> &nbsp;
>
> **C2: The evaluation criteria for RQ1 are too general, more details will be better to understand the changes or divergence in their personality, recall, and cognitive level. Some open-ended questions to provide details can be added to the evaluation.**
>
> **A2:**  We understand your concern that the evaluation criteria for RQ1 may initially appear too general and apologize for any confusion.
>
> To clarify, evaluators (both human and LLM-based) were provided with explicit persona descriptions for each patient prior to evaluation. These included details about personality traits, language proficiency, memory recall ability, and cognitive state. For example, a patient with a “verbose” persona was described as “Talkative, tends to speak at length, and provides highly detailed responses,” whereas CEFR A-level proficiency was described as “Beginner. Can make simple sentences.” Evaluators then rated the statements in Section 6.1.1 on a 4-point scale, with well-defined criteria for each score to ensure consistent and reliable assessments.
>
> While adding open-ended questions could have provided more nuanced insights, doing so would have placed an additional burden on evaluators. To improve efficiency and reduce costs, we did not systematically include open-ended questions in the study. Instead, when evaluators expressed dissatisfaction during consultations with PatientSim, they were asked to select a reason from the provided options. Additionally, evaluators were invited to highlight aspects of the interaction they found particularly challenging or impressive through a feedback survey.
>
> During the evaluation process, if physicians rated the realism of PatientSim with a low score, they were asked to select a reason from the following options (or specify their own): Insufficient emotional expression, Overly exaggerated emotional expression, Inconsistent behavior within the interaction, Behavior inconsistent with the assigned profile, Unnatural tone (overly formal or rigid), Mechanical repetition, Other (please specify).
>
> Out of 108 samples evaluated, only five were reported as disappointing. Four of these involved simulators with low language proficiency, and three of them also had low recall ability. Physicians indicated that these simulators provided limited and inadequate responses, most frequently selecting “Unnatural tone (overly formal or rigid)” and “Inconsistent behavior within the interaction” as reasons for dissatisfaction. Limited vocabulary resulting from low language proficiency was often perceived as an unnatural tone, while partial memory lapses were interpreted as inconsistent behavior. The remaining case was reported as having “Overly exaggerated emotional expression.”
>
> To gather additional feedback beyond the structured evaluation, we conducted wrap-up sessions and post-evaluation surveys. Physicians shared mixed impressions, for example:
> - “Interacting with an impatient persona reminded me of a real patient I saw yesterday at ED; the simulation was so realistic that it felt upsetting.”
> - “Sometimes the chatbot was unnecessarily irritable. Although this happens in real life, certain personalities were excessively irritable and impatient to the point of disrupting the interview. In practice, such patients would likely leave rather than continue the conversation.”
>
> Overall, evaluators were satisfied with the simulator’s performance, though some noted overly negative tendencies in a few samples. Feedback varied subtly depending on physicians’ individual clinical experiences. We will incorporate these findings into the final version of the manuscript.
>
> &nbsp;
>
> **C3: in Figure D11, how accurate for this binary classification for over 15+ different categories. As LLMs hallucinate, it will be good to test the robustness of such classifications for different LLMs and multiple times. Same for figure D12.**
>
> **A3:** To assess the accuracy and robustness of the binary classification in Figure D11 (and similarly D12), we manually annotated 411 sentences with binary labels as ground truth. These were then compared against LLM predictions to evaluate performance.
>
> To address potential LLM hallucinations and variability, we tested each task using two widely adopted LLM-as-judge models: GPT-4o and Gemini 2.5 Flash. The results, including sentence-level factual accuracy (as in D12), were benchmarked against human-labeled data. Based on comparative performance, Gemini 2.5 Flash was selected as the final evaluator for consistency across evaluations.
>
> This robustness analysis is summarized in Appendix F.4.1 and Table F15. For clarity, we will update the paper to explicitly link Section D.2.2 (where the classification results are shown) with Appendix F.4.1, to ensure the evaluation methodology and validation steps are easier to trace and understand.
>
> &nbsp;
>
> **C4: In Figure 20, the doctor and patient conversations all start with doctors who seems know the patient condition already. But in a realistic scenario, the first question from doctor could be why you are coming to ED?, then the conversation follows. Can you show a full conversation with several turns?**
>
> **A4:** Thank you for your comment, and we apologize for the confusion. To clarify, in our simulation setting, doctors begin the conversation without prior knowledge of the patient’s condition. Just as in real clinical encounters, the dialogue typically starts with open-ended questions such as “How can I help you today?” to elicit the patient's chief complaint and initiate the diagnostic process. Below is an example of such a multi-turn conversation:
>
> ```
> Doctor: Hello, how can I help you?
> Patient: I have been having a lot of pain in my abdomen. It started four days ago and it's getting worse. The pain is in my lower stomach and sometimes it goes to my back.
> Doctor: I'm sorry to hear that you're in pain. Can you describe the type of pain you're experiencing? Is it sharp, dull, cramping, or something else?
> Patient: The pain is dull, like an ache. It's not sharp, and it doesn't get better or worse with eating or walking. It's just a steady, dull feeling in my lower stomach.
> …
> Doctor: It's great to see you're mindful of your health. Do you have any significant health conditions in your family, such as heart disease, diabetes, or kidney problems?
> Patient: No, my family doesn't have any major health problems that I know of. My mom and dad are both alive and healthy, and my siblings don't have any serious conditions either.
> …
> Doctor: Thank you for that information. Based on everything you've shared, I will consider the following differential diagnoses:
> Kidney-related issues (such as obstruction or infection)
> Pancreatitis
> Gastrointestinal obstruction
> Hepatic complications (related to your liver)
> Musculoskeletal pain (possibly related to referred pain from the abdomen)
> ```
>
> In Figure 20 and the sample dialogue above, we presented only a partial snippet of a multi-turn conversation to ensure readability and to avoid disclosing sensitive, patient-specific information from the MIMIC-ED dataset, which has restricted access. The primary goal of this figure was to illustrate the expression of patient persona within the dialogue, rather than to depict the complete conversational flow.
>
> The full conversations including the doctor’s initial questions and all subsequent turns are available in our shared dataset:
> - `persona_test/expert_dialogue.jsonl`: contains full human-authored dialogues.
> - `*/llm_simulation/{llm backbone}/llm_dialogue.jsonl` : includes full dialogues generated by different LLMs.
>
> We will revise the figure caption to clarify this setup.

---

> > ### Comment · Reviewer_Fojh · 2025-08-05
> >
> > thanks for the examples and updates.

---

> > > ### Author Response · Authors · 2025-08-06
> > > **Sincerely expecting further discussions with Reviewer Fojh**
> > >
> > > Thank you for your reply and for taking the time to review our work. We would greatly appreciate it if you could let us know whether our previous response has fully addressed your concerns, or if you have any further questions or feedback.

---

> > > ### Author Response · Authors · 2025-08-08
> > > **Gentle follow-up regarding our previous response**
> > >
> > > Thank you again for your thoughtful feedback. As the discussion period is nearing its end, we would like to follow up to confirm whether our previous response fully addressed your concerns. If anything remains unclear or if you have further questions, we would welcome the opportunity to provide additional clarification. Please don’t hesitate to reach out.

---

### Official Review · Reviewer_CH3C · 2025-07-21

**Ethics Flags:** Data privacy, copyright, and consent
**Rating:** 4
**Confidence:** 3

**Summary:**

1.This paper introduces PATIENTSIM, a persona-driven patient simulator that generates realistic doctor-patient interactions by combining 170 clinical profiles from real-world MIMIC datasets and 37 unique patient personas defined across four axes (personality, language proficiency, medical history recall, and cognitive confusion).

2.Its primary contributions include a novel framework integrating clinical accuracy and diverse personas, comprehensive evaluation of eight LLMs (with Llama 3.3 identified as the top-performing open-source model, validated by clinicians), and an open-source, customizable, privacy-compliant platform serving as a testbed for medical dialogue systems and an educational tool.

3.The simulator addresses limitations of existing tools by enabling multi-turn, context-aware interactions that reflect real clinical dynamics, enhancing the training and evaluation of doctor LLMs and medical professionals.

**Dataset Code Accessibility:**

Yes

**Dataset Code Comments:**

1.For datasets: The submission uses real-world clinical data from MIMIC-IV, MIMIC-IV-ED, and MIMIC-IV-Note (publicly available under PhysioNet license) and constructs 170 structured patient profiles with 24 items, detailed in Appendix A. The derived dataset will be publicly available under the same license, ensuring accessibility and compliance. Documentation includes data sources, processing steps, and profile structure, reviewed by clinical experts.
2.For benchmarks: The paper provides detailed experimental settings (e.g., 170 profiles split into 108 for persona evaluation and 52 for factual accuracy/plausibility), evaluation metrics (Entail, ICov, ICon, plausibility), and clear procedures for LLM and human evaluation, supporting reproducibility.
3.For code and data: The authors state they will release experimental code via GitHub and share anonymized data/code during review, with a public version post-review. Appendix D details experimental settings, ensuring executability and full access to final forms.

**Ethical Comments:**

The work is limited by reliance on the MIMIC database (restricting generalizability, Sec. 8), inability to capture non-verbal cues (limiting persona depth, Sec. 8), and a small human evaluation sample (4 clinicians, Sec. 8). To improve, incorporating multimodal data (e.g., VR for non-verbal expressions) and expanding evaluator diversity could enhance realism and generalizability.

**Ethical Considerations:**

No, there are no or only very minor ethics concerns

**Limitations Weaknesses:**

The work is limited by reliance on the MIMIC database (restricting generalizability, Sec. 8), inability to capture non-verbal cues (limiting persona depth, Sec. 8), and a small human evaluation sample (4 clinicians, Sec. 8). To improve, incorporating multimodal data (e.g., VR for non-verbal expressions) and expanding evaluator diversity could enhance realism and generalizability.

**Strengths Contributions:**

PATIENTSIM's key strengths lie in integrating 170 real-world clinical profiles from MIMIC datasets and 37 diverse personas across four axes, validated by clinicians (avg. 3.89/4), filling gaps in prior simulators by merging clinical accuracy and persona diversity (Abstract, Sec. 7). It distinguishes itself from prior work by emphasizing multi-turn, context-aware interactions with nuanced traits, unlike single-turn or limited-persona alternatives (Sec. 2). The paper is well-organized, with informative figures (e.g., Fig. 1) and tables (e.g., Tables 1-4) clarifying its framework and results.

---

> ### Author Rebuttal · Authors · 2025-07-30
>
> We greatly appreciate your careful evaluation of our paper and your insightful feedback. We address your comment in detail below.
>
> **C1: The work is limited by reliance on the MIMIC database (restricting generalizability, Sec. 8), inability to capture non-verbal cues (limiting persona depth, Sec. 8), and a small human evaluation sample (4 clinicians, Sec. 8). To improve, incorporating multimodal data (e.g., VR for non-verbal expressions) and expanding evaluator diversity could enhance realism and generalizability.)**
>
> **A1:** We acknowledge this limitation, which is, to some extent, unavoidable given the nature of the study and the accessibility of available data. Simulating patients requires detailed clinical information, such as presenting symptoms, pain levels, and family medical history. These elements are critical for physicians during the initial diagnosis process, as they form the foundation of clinical reasoning and decision-making. Medical textbooks and clinical guidelines [1,2] emphasize the importance of collecting this information during the history-taking stage. Currently, the MIMIC database is the only publicly available dataset that provides this level of clinical detail. While this limits the generalizability of our findings, it reflects an unavoidable constraint given the need for clinically rich data to ensure simulation fidelity.
>
> Incorporating multimodal data to capture non-verbal cues, potentially through VR or other simulation technologies, is a promising direction for future work. However, this area remains relatively underexplored. As Steenstra et al. [3] note, current VR patient simulations often produce machine-like responses, lack natural intonation, and fail to represent personality traits in a meaningful way. Given these limitations, the use of VR in patient simulation may still be premature. Due to these constraints, most existing research on simulating patient-doctor interactions continues to rely on text-based simulations. Within this constraint, our patient simulator uses tone indicators and punctuation, such as ellipses or emphatic phrasing, to express subtle emotional cues and enhance the realism of the simulation. For example, a line like “Look, I’ve had a lot, can’t you see I’m tired?” reflects an impatient tone, while “I was sitting, then… I don’t know. I just fell.” expresses confusion and hesitation (Figure F20).
>
> Regarding the human evaluation, we were limited to four clinicians due to budget constraints, with a total of €2,500 allocated for 45 hours of evaluation work. Despite the small sample size, we consider the evaluators’ insights to be highly valuable given their substantial clinical expertise. Two of the evaluators are licensed physicians with four years of clinical experience. The other two are also licensed physicians with six years of clinical experience and additionally hold nursing licenses, with 13 and 17 years of prior nursing practice, respectively. The clinicians were recruited via Ingedata, an AI data annotation company that serves as an intermediary between clinicians and research teams. Importantly, all four evaluators are affiliated with different hospitals or medical institutions, contributing to a wider range of clinical perspectives and helping to mitigate potential institutional bias.
> Notably, many prior studies in this domain have relied solely on LLM-based evaluations [4,5] or involved a smaller number of human evaluators, often medical students [6,7]. This is partly due to the difficulty and cost of recruiting experienced medical professionals. In this context, and within the given constraints, our approach provides a robust and realistic human evaluation framework, grounded in clinical expertise and institutional diversity.
>
> While certain limitations are inevitable due to practical, technical, and financial constraints, we believe our study makes a concrete and meaningful contribution. As technologies and data accessibility continue to advance, we see our work as laying the groundwork for more immersive and sophisticated simulation research in the future.
>
> [1] E. C. Toy, B. Simon, K. Takenaka, T. H. Liu, and A. J. Rosh. Case Files Emergency Medicine. McGraw-Hill Education / Medical, New York, 4th edition, 2017. ISBN 9781259640827.
> [2] Osmosis. OLD CARTS: History-taking Mnemonic. n.d. Available at: https://www.osmosis.org/answers/old-carts-history-taking-mnemonic. Accessed: July 27, 2025.
> [3] Steenstra et al., Scaffolding Empathy: Training Counselors with Simulated Patients and Utterance-level Performance Visualizations, In CHI '25, 2025.
> [4] H. Liu et al., Med-pmc: Medical personalized multi-modal consultation with a proactive ask-first-observe-next paradigm, ArXiv, 2024.
> [5] H. Wei et al., Medco: Medical education copilots based on a multi-agent framework, ECCV 2024 Workshops, 2024.
> [6] Y. Liao et al., Automatic interactive evaluation for large language models with state aware patient simulator, ArXiv, 2024.
> [7] Z. Fan et al., Ai hospital: Benchmarking large language models in a multi-agent medical interaction simulator. In COLING, 2025.

---

> > ### Author Response · Authors · 2025-08-07
> > **Sincerely looking forward to further discussions with Reviewer CH3C**
> >
> > Thank you for your thoughtful review of our paper. As the discussion period is nearing its end, we would sincerely appreciate it if you could let us know whether our response has fully addressed your concerns. Please let us know if you have any further questions or feedback.

---

> > ### Author Response · Authors · 2025-08-09
> > **Gentle follow-up regarding our previous response**
> >
> > As the discussion period nears its conclusion, we would like to kindly follow up to confirm whether our rebuttal has addressed your concerns. Please let us know if any questions remain, as we would be pleased to provide further clarification. Thank you for your time and consideration.

---

### Note · Authors · 2025-08-12

We sincerely thank the ACs for their oversight and the reviewers for their time and constructive feedback. To aid the final decision, we restate our contributions and address the main concerns.

We introduce PATIENTSIM, a patient simulator for generating realistic, diverse clinical personas grounded in medical expertise. Realistic patient interaction is essential for training and evaluating doctor LLMs, yet existing simulators lack the diversity observed in practice. PATIENTSIM addresses this gap by combining: (1) 37 personas across four axes (personality, language proficiency, history recall, and cognitive confusion), and (2) clinical profiles from real datasets (MIMIC-ED, MIMIC-IV). LLM and human evaluations confirm its value as a robust, reproducible testbed for medical dialogue systems.

Reviewer Concerns & Responses:
1. Text-Based Simulation: While multimodal or VR simulations could further enhance realism, such methods are not yet mature for reliable clinical use. Given these limitations, we adopted a text-based format, which remains the standard in this domain.

2. Dataset Generalizability: We acknowledge limitations but note that MIMIC databases are the only publicly available datasets with detailed, structured clinical data. Its use is standard in medical AI, enabling reproducibility and transparency that are not possible with proprietary datasets.

3. Human Evaluation Size: Budget limits allowed four licensed clinicians (4–6 years’ experience; two also with 13–17 years’ prior nursing), totaling 45 hours (€2,500). They brought diverse institutional backgrounds. Many prior works rely solely on LLM evaluation or fewer/less experienced human evaluators. Within constraints, our approach offers a robust, clinically grounded evaluation.

4. Scope - Single-Session History-Taking & Five Diseases: The focus reflects both dataset constraints and clinical feasibility. Only 2–3% of ED visits in MIMIC include usable pre-intervention imaging/lab data, making broader simulation impractical. We selected five prevalent ED diseases with distinctive symptom profiles detectable via history-taking alone and with sufficient sample sizes, as advised by two medical experts.

We value the reviewers’ insights, but note that some expectations exceed current data availability and technical capabilities. Our design choices reflect pragmatic, clinically informed decisions aimed at advancing realistic, diverse, and reproducible evaluation in medical dialogue research.

---

### Decision · Program_Chairs · 2025-09-18

**Decision:**

Accept (spotlight)

**Comment:**

This paper introduces PATIENTSIM, a persona-driven patient simulator designed to generate realistic doctor-patient interactions. The system integrates 170 clinical profiles derived from MIMIC-IV and MIMIC-ED datasets with 37 distinct patient personas, modeled along four axes: personality, language proficiency, medical history recall, and cognitive confusion. Eight LLMs were evaluated, with Llama 3.3 selected as the final backbone model. The simulator achieves an average quality score of 3.89/4 across six evaluation criteria.

Strengths
- Incorporates diverse, multi-dimensional patient personas that capture real-world consultation complexities.
- Goes beyond prior work by modeling communication challenges (e.g., personality, recall, confusion) rather than restricting to simplified patient roles.
- Employs a mixed evaluation framework combining automated metrics and expert human judgment.

Weaknesses
- Reliance on the MIMIC database may limit the diversity of clinical presentations.
- Scope restricted to five diseases, single ED visits, with only textual (non-multimodal) patient information.
- Clinical validation involved only four general practitioners.

Despite these limitations, the reviewers agree that this paper presents a well-motivated and timely contribution. It addresses an important gap in the evaluation of medical AI systems by offering a richer, persona-driven simulation framework.
The authors provided thorough clarifications during the rebuttal phase.
No ethics concerns were identified.

===== FINAL UPDATE FROM DB Track PCs ====

The final decision for this paper has been taken by the program chairs after consultation with the SACs. All Senior Area Chairs have ranked papers according to the feedback from the AC during the review process. We decided to leave the original meta-review to reflect the opinion of the AC in light of the initial discussions with reviewers and SAC.